# ZoomVLM: A Tuning-Free Framework for Efficient Video Understanding via Adaptive Zooming in Vision-Language Models

## Abstract

Recent advances in vision-language models (VLMs) have led to impressive progress in video understanding. However, despite their promising performance, existing state-of-the-art (SOTA) solutions require an excessive number of tokens (e.g., up to 6,272 tokens in the Llava-OneVision model) to represent input videos, leading to a non-negligible bottleneck in inference efficiency. Motivated by findings in human perception, where individuals first focus on high-level overviews and then zoom into specific areas for detailed information, we hypothesize that a similar approach can enhance the inference efficiency of VLMs by reducing the number of tokens needed to represent videos. Based on this hypothesis, we propose ZoomVLM, a tuning-free, plug-and-play efficient video processing framework for video VLMs. ZoomVLM first generates an overview of the entire video and then adaptively zooms in and out on different parts based on the content being generated. Our key insight is that the attention distributions in the Large Language Model (LLM) within the VLM can provide sensible guidance on where to focus (by allocating more tokens) and where to discard (by dropping tokens) during inference. Specifically, ZoomVLM integrates two key components: (1) a Video Overview Augmenter, which enables cost-effective high-level understanding by augmenting downsampled video overview with a few high-resolution keyframes; and (2) an Adaptive Token Adjustment, which predicts the importance of different video parts in the upcoming generation process and adjusts the number of tokens allocated to each part according to their importance. Extensive experiments and ablation studies across two challenging open-ended video understanding benchmarks and four models validate that ZoomVLM effectively improves inference efficiency by reducing the number of tokens and boosting throughput in terms of the number of generated tokens per second without degradation in achievable accuracy. Specifically, when applying ZoomVLM to Llava-Next-Video-7B-DPO, ZoomVLM achieves a 30% higher token generation rate with a 0.259 improvement in the Video Detail Description score.

## 1 Introduction

Leveraging pretrained Vision Transformers (ViTs) (Dosovitskiy et al., 2021; Liu et al., 2021; Cai et al., 2023; Tu et al., 2022; Liu et al., 2022b;a) and Large Language Models (LLMs) (OpenAI, 2023a; Brown et al., 2020; OpenAI, 2023b; Touvron et al., 2023a;b; Dubey et al., 2024), recent vision-language models (VLMs) have revolutionized video understanding by seamlessly integrating visual and textual information (Li et al., 2024a; Liu et al., 2023b; 2024a; 2023a; Zhang et al., 2024b; Lin et al., 2024). These models have achieved state-of-the-art (SOTA) performance in video understanding tasks, underscoring a significant stride towards automating complex interpretative tasks that require an intricate understanding of both visual elements and narrative contexts, such as robotics Hu et al. (2023); Zhang et al. (2024a); Brohan et al. (2023), augmented reality and virtual reality (AR/VR) assistants Konenkov et al. (2024); Bi et al. (2023), and autonomous vehicles Tian et al. (2024); Renz et al. (2024).

Despite exciting advancements, the efficiency bottleneck of video VLMs has become a new limitation hindering their application to a broader range of applications. This is because SOTA VLMs typically

use pretrained ViTs to encode video frames into tokens, which are then concatenated into lengthy sequences and fed into LLMs alongside text instructions (Zhang et al., 2024b; Li et al., 2024c;a). Although effective, this encoding approach results in extremely long inputs, for example, one of the SOTA video VLMs, Llava-OneVision (Li et al., 2024a), requires up to 6,272 tokens to represent a single video, as shown in Fig. 1 (a), posing substantial efficiency challenges compared to image VLMs Liu et al. (2023b); Lin et al. (2024); Li et al. (2022; 2023) and conventional LLMs Touvron et al. (2023a;b); Dubey et al. (2024) due to the increased input token length to the LLM. Moreover, as illustrated in Fig. 1 (b), the number of video tokens plays a critical role in the performance of VLMs. Naively reducing the number of tokens through uniform downsampling and pooling, as adopted in existing works Li et al. (2024a); Zhang et al. (2024b), results in a non-neglectable drop of up to 0.2 in the video description score on the Video Detail Description benchmark Maaz et al. (2023), further intensifying the challenge of improving accuracy and efficiency trade-off in video VLMs.

Contrary to current VLMs that uniformly process video data, human cognition employs a more strategic approach to interpretation, beginning with a broad assessment followed by selective attention (Cherry, 1953; Lachter et al., 2004; Sternberg & Sternberg, 2006). Initially, humans skim the video to capture a high-level overview, establishing a contextual framework for the entire scene. This global perspective then informs selective focus on specific parts of the video to extract detailed information relevant to the inquiry. By combining both the high-level context and localized details, humans efficiently and accurately address complex questions about the video content, thereby eliminating the need to retain every video detail.

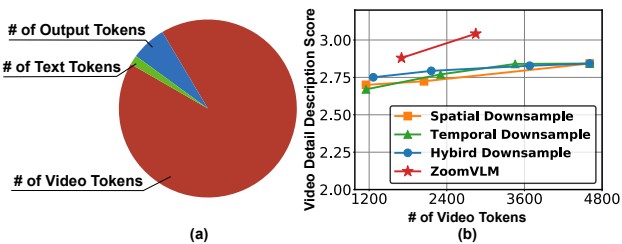

Figure 1: (a) Breakdown of input token counts in the Video Detail Description benchmark (Zhang et al., 2024b); and (b) the trade-off between evaluation score ↑ and the number of tokens when using naive downsampling from the spatial, temporal, and both dimensions of a video as commonly adopted in SOTA solutions Xu et al. (2024b); Li et al. (2024a); Zhang et al. (2024b) (detailed experiment settings can be found in App. G) is compared to our proposed ZoomVLM, applied on the pretrained Llava-Next-7B-DPO (Li et al., 2024a), at the Video Detail Description benchmark (Zhang et al., 2024b).

Inspired by the above human perception strategy, we aim to design a "zoom-then-focus" approach for video VLMs to optimize the trade-offs between accuracy and efficiency by emulating human perceptual processes. Specifically, we aim to first present video VLMs with a compact representation of the video that offers a high-level overview and can be encoded with a relatively small set of tokens. Then, during the generation process, the model can selectively zoom into video segments that are crucial for resolving the specific inquiry. The key insight of our approach is the attention distribution from LLMs within VLMs can precisely identify areas that need to be zoomed in during inference, thus ensuring efficient and accurate video understanding without an excessive efficiency bottleneck from the number of video tokens needed.

Our contributions in this paper can be summarized as follows:

- Drawing inspiration from human behavior in perception, we propose a tuning-free, plug-and-play efficient video processing pipeline for VLMs, dubbed ZoomVLM, that first generates an overview of the entire video and then adaptively zooms in and out on different parts of the video based on the content being generated. By adaptively allocating video tokens to different parts of the video, ZoomVLM addresses the primary efficiency bottleneck in existing video VLMs, while ensuring all essential information needed to understand the video and generate responses is preserved.
- ZoomVLM integrates two key components to efficiently select necessary information by leveraging the attention distribution within the VLM: (1) a Video Overview Augmenter, which creates an informative video summary by augmenting downsampled video with high-resolution keyframes to capture crucial details; and (2) an Adaptive Token Adjustment, which predicts the significance of different video parts in the generation process and then adjusts token allocation accordingly.
- Extensive experiments and ablation studies across two challenging open-ended video understanding datasets, and four models validate that ZoomVLM effectively improves inference

efficiency by reducing the number of tokens and boosting throughput in terms of the number of generated tokens per second without degradation in achievable accuracy. Specifically, when applying ZoomVLM to Llava-Next-Video-7B-DPO (Li et al., 2024a), ZoomVLM achieves a 30% higher token generation rate with a 0.259 improvement in the Video Detail Description score.

# 2 RELATED WORKS

## 2.1 VLMS AND VIDEO VLMS

VLMs bridge visual and textual understanding by converting images and videos into tokens for joint reasoning (Liu et al., 2024a; Zhang et al., 2024b; Lin et al., 2024). Image-based VLMs (Liu et al., 2023b; Li et al., 2022; Zhang et al., 2023c) achieve impressive results using large-scale image-text datasets. Extending to video data introduces challenges due to temporal complexity, leading to two main types of video VLMs. **Type 1: Sequentially processing frames as tokens.** Models like Liu et al. (2024a); Zhang et al. (2023a) adapt image-based VLMs by feeding multiple frames sequentially, treating frames as a large image. This achieves state-of-the-art video understanding but requires an enormous number of tokens—up to 6,272 (Li et al., 2024a)—leading to efficiency bottlenecks. **Type 2: Summarizing videos into a fixed number of tokens.** These models use fewer tokens before feeding them into LLMs. For example, Li et al. (2023); Lin et al. (2024) employ the Query Transformer (Q-Former)(Li et al., 2023) with learnable query vectors to extract visual features. This reduces computational overhead but can result in accuracy loss, as crucial details may be lost.

Some works try to balance efficiency and accuracy but require drastic architectural changes, making them hard to apply to pretrained VLMs. For instance, Feichtenhofer et al. (2019); Xu et al. (2024a) use "slow" and "fast" pathways, and Weng et al. (2024) introduce efficient attention mechanisms, but both alter the VLM structure. In contrast, our proposed ZoomVLM offers a **tuning-free, plug-and-play** framework that enhances efficiency without compromising accuracy, inspired by human "zoom-then-focus" strategies.

## 2.2 ATTENTION PATTERNS IN TRANSFORMERS

With the application of LLMs to various domains, understanding their attention mechanisms is evolving. Early studies (Clark et al., 2019; Vig, 2019; Sun & Lu, 2020) analyzed attention patterns in small-scale transformers to interpret model behavior. More recent works reveal how attention distributions provide insights into LLM processing. For example,Xiao et al. (2023) observe attention sinks, where certain tokens receive disproportionately large attention scores, affecting performance in streaming applications. Similarly,Yu et al. (2024) find that not all attention sinks positively impact accuracy. Additionally,Sun et al. (2024) identify massive activations leading to concentrated attention probabilities and implicit biases. To address these issues,Zhang et al. (2023b) introduce a method to steer LLM attention post-hoc, enhancing instruction-following by manipulating attention distributions without altering model parameters.

However, these works primarily focus on attention in LLMs rather than VLMs. While attention is crucial for aligning visual and textual modalities in VLMs, it remains underexplored. Our proposed method, ZoomVLM, addresses this gap by leveraging attention patterns within the LLM component of VLMs to guide efficient video processing. By analyzing attention distributions, ZoomVLM adaptively allocates computational resources to different video segments, improving inference efficiency without sacrificing accuracy.

## 2.3 KV CACHE COMPRESSION FOR LONG SEQUENCES

With the increasing demand for longer context inputs in LLMs, the large size of cached key-value tokens (KV cache), which store attention keys and values during generation to prevent re-computation, has become a major bottleneck in the generation stage in terms of latency and memory usage (Yuan et al., 2024). To compress the KV cache and better handle long input sequences, two types of compression solutions have emerged: (1) Token dropping: (Xiao et al., 2023; Zhang et al., 2024d; Xiao et al., 2024; Wang et al., 2024) propose dropping some tokens from the KV cache, similar to pruning weights in neural networks; (2) Token quantization: (Sheng et al., 2023; Liu et al., 2024b) adopt a quantization approach for storing the KV cache, reducing data loading time, which is critical

for memory-bottlenecked LLM inference. Some recent works also explore combining these two approaches (Zhang et al., 2024c).

However, these methods have two main drawbacks: (1) Their effectiveness has not been verified in VLMs, which are more complex than LLMs due to the requirement of understanding both visual and textual information; (2) They struggle to handle the dynamic information needs during generation based on input. In contrast, our proposed ZoomVLM pioneers the study of token cache compression in VLMs, while developing a compression strategy that dynamically adapts to the latest information received during generation.

## 3 PRELIMINARY ON VIDEO VLMS

Current SOTA video VLMs are developed through a hybrid model design that leverages the strengths of distinct pretrained models to process complex video and text inputs. In particular, a current VLM usually integrates a pretrained ViT ($\mathcal{M}_V$) to encode each video frame into a series of tokens, a projection layer ($\mathcal{M}_P$) to align these token embeddings with the embedding space of a pretrained LLM ($\mathcal{M}_L$), and the LLM itself to synthesize and generate responses based on the combined video-text information in an autoregressive manner. For an input video with $N$ frames, denoted as $\mathcal{V} = \{\mathbf{V}^0, \cdots, \mathbf{V}^{N-1}\}$, where each $\mathbf{V}^i \in \mathbb{R}^{H \times W}$ having a spatial dimension of $H \times W$ and a corresponding text prompt $\mathbf{T}$, the processing pipeline can be described as follows:

$$\mathbf{P} = \text{Concat}([\mathcal{M}_P(\mathbf{V}^i) \text{ for } i = 0, \cdots, N-1]), \tag{1}$$

$$\mathbf{x}_1 = \text{argmax}(\mathcal{M}_L(\mathbf{x}_0 \mid [\mathbf{T}, \mathbf{P}])), \tag{2}$$

$$\mathbf{x}_t = \text{argmax}(\mathcal{M}_L(\mathbf{x}_{t-1} \mid [\mathbf{T}, \mathbf{P}, \mathbf{x}_0, \cdots, \mathbf{x}_{t-2}])), \tag{3}$$

where $\mathbf{P}$ denotes the concatenated token embeddings from all video frames, $\mathbf{x}_0$ is the special token used to start the generation process, each $\mathbf{x}_t$ represents the output token at generation step $t \in [1, \cdots, T]$, and $T$ is the total number of generation steps. Eq. 2 represents the first generation step, while Eq. 3 describes the subsequent step in the generation process.

**Attention Mechanism in Autoregressive Generation.** In each step of the autoregressive generation, the attention mechanism plays a critical role in determining the influence of previously generated tokens on the current output. At step $t$, the model computes the attention scores $\alpha_{t,i}$ for all previous tokens, including $\mathbf{P}$, $\mathbf{T}$, and $\mathbf{x}_0, \cdots, \mathbf{x}_{t-1}$. The attention scores $\alpha_{t,i}$ for each token at step $t$ are computed as:

$$\alpha_{t,i} = \frac{\exp(\mathbf{q}_t^\top \mathbf{k}_i)}{\sum_{j=0}^{t-1} \exp(\mathbf{q}_t^\top \mathbf{k}_j)}, \tag{4}$$

where $\mathbf{q}_t$ is the query and $\mathbf{k}_i$ is the key vector of the $i$-th token. The output $\mathbf{h}_t$ of attention is then determined using these scores and the value vectors $\mathbf{v}_i$:

$$\mathbf{h}_t = \sum_{i=0}^{t-1} \alpha_{t,i} \mathbf{v}_i, \tag{5}$$

**Efficiency Bottleneck in Autoregressive Generation.** During the generation process with long input sequences, such as those involving thousands of tokens from $\mathbf{P}$, the efficiency is initially constrained by the computational demands required for the prefilling generation step (i.e., Eq. 2). However, in the later autoregressive generation steps, memory overhead becomes the major efficiency bottleneck due to the need to manage a large key-value (KV) cache. This cache, which accumulates tokens generated by $[\mathbf{T}, \mathbf{P}, \mathbf{x}_0, \cdots, \mathbf{x}_{t-2}]$ (as in Eqs. 3, 4, and 5), must be loaded from the GPU device memory for each subsequent generation step (Liu et al., 2024b). Consequently, minimizing the number of tokens in $[\mathbf{T}, \mathbf{P}]$ becomes crucial for enhancing the overall generation speed of VLMs.

## 4 ZOOMVLM FOR EFFICIENT VIDEO VLM INFERENCE

### 4.1 OVERVIEW

As discussed in Sec. 3, our objective is to enhance the inference efficiency of video VLMs during the autoregressive generation process by reducing the number of video tokens (i.e., $\mathbf{P}$ in Eq. 3) that need to be processed and stored in the KV cache of LLMs. Drawing inspiration from human

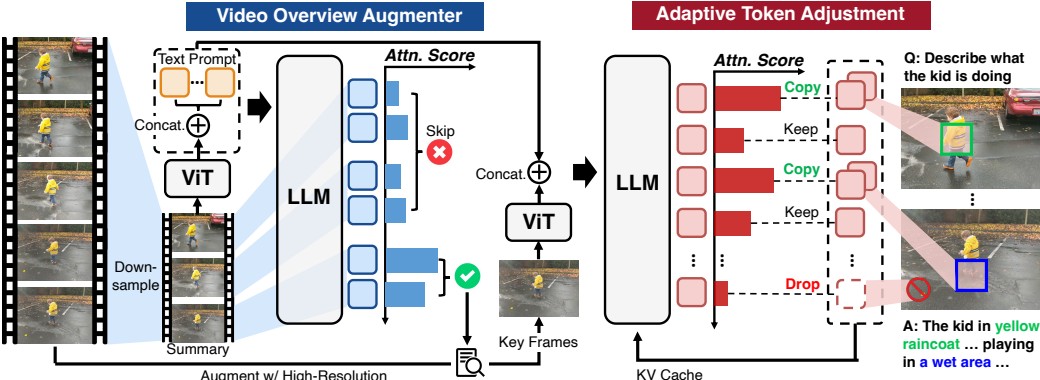

Figure 2: An overview of the proposed ZoomVLM framework. ZoomVLM incorporates two key components: the Video Overview Augmenter (in Sec. 4.2), which generates a concise and informative video overview enriched with key high-resolution frames, and the Adaptive Token Adjustment module (in Sec. 4.3, which enables the video VLM to focus on essential elements while filtering out irrelevant ones by selectively copying or dropping corresponding tokens.

perception, as illustrated in Fig. 2, ZoomVLM incorporates two key components: (1) the Video Overview Augmenter, which first spatially downsamples the video to reduce the number of tokens and then uses the LLM's attention map to select keyframes essential for the entire generation process, and (2) the Adaptive Token Adjustment module, which predicts token importance during generation, allowing the model to dynamically zoom in on or zoom out from different video segments, efficiently retaining relevant information with minimal additional overhead. In the remainder of this section, we introduce the two key components in detail, with the Video Overview Augmenter and the Adaptive Token Adjustment module described in Sec. 4.2 and Sec. 4.3, respectively.

### 4.2 VIDEO OVERVIEW AUGMENTER

**Motivation and Goal.** In the Video Overview Augmenter, our objective is to provide the VLM with a compact yet informative representation of the video, enabling effective understanding. However, as illustrated in Fig 1, simply reducing the number of tokens via common techniques such as spatial pooling or temporal uniform sampling (Li et al., 2024a; Zhang et al., 2024b; Xu et al., 2024b) leads to significant accuracy degradation. This drop in accuracy occurs due to the loss of critical detailed information in the compressed video overview.

To address this issue, in Video Overview Augmenter, we aim to design an approach that can effectively provide the compressed overview video with necessary detailed information. The key challenge is identifying an accurate and efficient mechanism to locate this critical information during the generation process, thereby enhancing the video overview and improving the VLM's understanding and task performance. In the remainder of this section, we first present our analysis of existing works to identify the source of potential mechanism we can leverage as the identifier, followed by a detailed introduction of our proposed Video Overview Augmenter.

**Analysis of Existing Works on Attention Mechanism.** Attention mechanism in large transformer models effectively reveal the model's focus during inference. For example, in ViTs, excessive attention to specific locations can indicate overfitting Yu et al. (2023), while in LLMs, attention enhances instruction-following capabilities Zhang et al. (2023b). Building upon this interpretability, we leverage attention to identify important parts of the video conditioned not only on the text prompt (e.g., question), as in prior works Zhang et al. (2023b), but also on the content generated thus far. This dual conditioning provides additional guidance to better assess the relevance of different video segments during the generation process. By eliminating irrelevant information that is generally considered important but unrelated to the generated content, we reduce redundancy in $\mathbf{P}$ as in Eq. 1 and enhance the model's efficiency and performance.

**Proposed Approach.** The proposed Video Overview Augmenter leverages a condensed overview of the video to identify important frames and augment the video's representation with those frames, enhancing text generation, as shown in Fig. 2. This process begins by generating a high-level summary of the video through spatial pooling and temporal sampling, effectively reducing the video's size and the number of tokens required for its representation. This condensed overview is then input into

the VLM to initiate text generation. After generating a predefined number of tokens, we assume the model has gathered sufficient context to identify the most critical segments of the video to facilitate the remaining generation process. We leverage the attention distribution from the LLM to select a small set of video frames with the highest frame-wise average attention scores as critical segments. We enhance the resolution of these selected frames and process them through the ViT to extract detailed information. The augmented video tokens are then fed back into the LLM, allowing the generation process to proceed with improved context and specificity.

This process can be formally defined as follows. Given the input video $\mathcal{V}$, we first perform spatial and temporal sampling to reduce its overall size, generating $\hat{\mathcal{V}} = \{\hat{\mathbf{V}}^0, \cdots, \hat{\mathbf{V}}^{\lfloor \frac{N-1}{r} \rfloor}\}$, where $r > 1$ is the temporal uniform sampling ratio and $\lfloor . \rfloor$ denotes the flooring operation. Each $\hat{\mathbf{V}}^i \in \mathbb{R}^{\hat{H} \times \hat{W}}$ is the spatially downsampled version of frame $\mathbf{V}^{\mathbf{i}}$, with $H > \hat{H}$ and $W > \hat{W}$.

Next, we obtain the token representation $\hat{\mathbf{P}}$ of the overview video using

$$\hat{\mathbf{P}} = \text{Concat}([\mathcal{M}_P(\hat{\mathbf{V}}^i) \text{ for } i = 0 \ldots \lfloor \frac{N-1}{r} \rfloor]), \tag{6}$$

and start generating a short sequence consisting of $s$ tokens following Eqs. 2, and 3. By step $s$, we consider the LLM has sufficient information to identify a set of $k$ critical frames $\mathcal{A} \subset \hat{\mathcal{V}}$. To construct $\mathcal{A}$, we first select a fixed subset of layers $\mathbf{L} \subset \{0, 1, \cdots, L-1\}$ from all $L$ layers of the LLM. We empirically observe that selecting $\mathbf{L}$ from the middle part of the LLM exhibits better alignment between their identified frame-wise importance and human expectations, as visualized and analyzed in Sec. 5.3. We then construct $\mathcal{A}$ as follows:

$$A_i = \frac{1}{|\mathbf{L}| \times T_P} \sum_{l \in \mathbf{L}} \sum_{j=0}^{T_P - 1} \alpha_{s,j}^{(l)}, \tag{7}$$

$$\mathcal{A} = \{\hat{\mathbf{V}}^i \mid A_i \text{ is among the top } k \text{ scores}\}, \tag{8}$$

where $A_i$ is the average attention score for frame $\hat{\mathbf{V}}^i$ at step $s$, $T_P$ is the number of token to represent each $\hat{\mathbf{V}} \in \hat{\mathcal{V}}$, and $\alpha_{s,j}^{(l)}$ is the attention score at layer $l$ and generation step $t$ for token $j$.

Finally, we augment the video overview with collected critical frames. Specifically, for each $\hat{\mathbf{V}}^i \in \mathcal{A}$, we retrieve the original high-resolution frame $\mathbf{V}^i$ and obtain its token representation $\mathbf{P}_C$ using Eq. 1. We then concatenate $\hat{\mathbf{P}}$ with $\mathbf{P}_C$ and resume the generation process as follows:

$$\mathbf{x}_t = \text{argmax}\left(\mathcal{M}_L\left(\mathbf{x}_{t-1} \mid [\mathbf{T}, \hat{\mathbf{P}}, \mathbf{P}_C, \mathbf{x}_0, \ldots, \mathbf{x}_{t-2}]\right)\right), \quad \text{for } t = s+1, \ldots, T, \tag{9}$$

where $T$ is the total number of generation steps.

### 4.3 ADAPTIVE TOKEN ADJUSTMENT

**Motivation and Goal.** Although the Video Overview Augmenter provides ZoomVLM with a cost-effective overview containing most of the necessary information for generation, we observe that answers to questions about a provided video typically adaptively focus on different locations within the video. This behavior aligns with human perception processes as indicated in (Sternberg & Sternberg, 2006; Lachter et al., 2004; Cherry, 1953). However, existing video VLMs (Li et al., 2024a; Lin et al., 2024; Li et al., 2024c; Zhang et al., 2024b) and the augmented overview information from the Video Overview Augmenter consider the video representation as static, creating a mismatch between the LLM's perception process and the generation process. To alleviate this issue, in Adaptive Token Adjustment, we aim to develop a low-cost approach that adjusts the video representation during the generation process. By emphasizing information critical for the current generation step and mitigating the influence of unrelated information, Adaptive Token Adjustment aims to better align the adjusted video representation at each generation step with the content being generated, thereby improving generation accuracy.

**Proposed Approach.** To efficiently identify which parts of the video should be emphasized or ignored with minimal computational overhead, we leverage attention as a low-cost identifier. Specifically, following the Video Overview Augmenter, we periodically activate Adaptive Token Adjustment by examining the attention distribution across $\mathbf{L}$ layers in the LLM. Tokens receiving high attention are emphasized by duplicating them, while tokens with low attention are discarded to reduce redundancy.

Formally, we define:

$$\mathcal{E} = \{j \mid \alpha_{t_A,j} \text{ is among the top } m \text{ highest scores}\}, \tag{10}$$

$$\mathcal{I} = \{j \mid \alpha_{t_A,j} \text{ is among the top } m \text{ lowest scores}\}. \tag{11}$$

The adjusted token sequence $\mathbf{P}'$ is constructed by duplicating tokens in $\mathcal{E}$ and removing tokens in $\mathcal{I}$:

$$\mathbf{P}' = \text{Concat} \left( \left\{ \begin{array}{ll} [\mathbf{p}_j, \mathbf{p}_j], & \text{if } j \in \mathcal{E}, \\ \mathbf{p}_j, & \text{if } j \notin \mathcal{I}. \end{array} \right) \right. \tag{12}$$

where $\mathbf{p}_j$ is the embedding of the $j$-th video token. We then continue the generation using $\mathbf{P}'$:

$$\mathbf{x}_t = \text{argmax} \left( \mathcal{M}_L \left( \mathbf{x}_{t-1} \mid [\mathbf{T}, \mathbf{P}', \mathbf{x}_0, \ldots, \mathbf{x}_{t-2}] \right) \right), \quad \text{for } t = t_A + 1, \ldots, T. \tag{13}$$

## 5 EXPERIMENT RESULTS

### 5.1 EVALUATION SETTINGS

**Models, Tasks, and Datasets.** We evaluate ZoomVLM across four SOTA video VLMs including Llava-Next-Video-7B-DPO (Li et al., 2024a), Llava-Next-Video-7B (Li et al., 2024a), Llava-Next-Interleave-7B-DPO (Li et al., 2024c), and Llava-OneVision-0.5B-OV (Li et al., 2024a). The evaluation is conducted on challenging video understanding tasks across three benchmarks: Video Detail Description (VDD) (Zhang et al., 2024b) and Video-ChatGPT (Maaz et al., 2023). This comprehensive assessment aims to evaluate ZoomVLM's performance across diverse and complex settings.

**Evaluation Settings.** We conduct the evaluation primarily following the default settings and metrics of each benchmark and model. Specifically, we adopt the settings from (Li et al., 2024b) for VDD and Video-ChatGPT. During inference, we set the default video resolution to $H = W = 336$ with 32 frames. We construct $\mathbf{L}$ to include layers 5–20 for the Llava-Next series models and layers 10–25 for Llava-OneVision, considering that their backbones have different numbers of layers. In the Video Overview Augmenter, we set the initial generation step $s = 20$, the number of selected frames $k = 10$, the temporal downsampling ratio $r = 1.5$, and the spatial resolution of the condensed video $\hat{H} = \hat{W} = 224$. During Adaptive Token Adjustment, we perform adjustments periodically every three sentences and set $m = 10$. All experiments are run on a single A100-80GB GPU with a batch size of 1.

**Evaluation Metrics.** Following existing works, we use GPT models (Brown et al., 2020; OpenAI, 2023b) to score the generated responses to evaluate the generation quality. Specifically: (1) For VDD, we employ GPT-3.5 to compare the differences between the generated responses and reference descriptions. (2) For Video-ChatGPT, GPT-3.5 assesses the responses based on five aspects: Correctness of Information (Correctness), Detail Orientation (Detail), Contextual Understanding (Context), Temporal Understanding (Temporal), and Consistency. To evaluate the efficiency, we use the number of generated tokens per second (Token/sec) to evaluate the generation throughput and number of video tokens needed (# Video Tokens) to represent the theoretical improvement in representing the information in videos efficiently.

**Baselines.** We compare ZoomVLM with two baselines including the vanilla model and Slow-Fast Llava (Xu et al., 2024b), which is the pioneering work to reduce the number of video tokens in the SOTA Llave-series models in a tuning-free manner.

### 5.2 BENCHMARKING ZOOMVLM ON VIDEO UNDERSTANDING BENCHMARKS

We begin by benchmarking ZoomVLM by integrating it into SOTA video VLMs to validate its capability of improving the accuracy-efficiency trade-off on complex, open-ended video understanding tasks. As shown in Table 1, ZoomVLM achieves up to a 0.260 higher score on VDD and a 0.432 higher average score on Video-ChatGPT, along with a 22%∼30% increase in generation throughput.

Compared to the SlowFast baseline (Xu et al., 2024b), ZoomVLM consistently attains a 0.012 to 0.344 higher score on VDD and a 0.110 to 0.319 higher average score on Video-ChatGPT, while reducing the number of video tokens by 22.6% to 33.6%. Although SlowFast also uses uniform spatial and temporal downsampling, the significant improvement achieved by ZoomVLM indicates that uniformly processing all video parts is suboptimal. Techniques like our Video Overview Augmenter, which selectively augments the video overview based on the input video, text prompt, and initial generated

Table 1: Benchmark ZoomVLM on its capability to improve the accuracy-efficiency trade-off of SOTA video VLMs on the video understanding task.

| Model | Method | Video-ChatGPT | | | | | | VDD | Efficiency | |
| --- | --- | --- | --- | --- | --- | --- | --- | --- | --- | --- |
| | | Correctness | Detail | Context | Temporal | Consistency | Avg. | Score | Token/sec | # Video Tokens |
| Llava-Next-Video-7B-DPO | Vanilla | 3.094 | 2.601 | 3.511 | 2.313 | 3.186 | 2.941 | 2.843 | 33 | 4608 |
| | SlowFast | 3.396 | 2.878 | 3.651 | 2.708 | 3.513 | 3.229 | 2.864 | 32 | 3680 |
| | Ours | **3.529** | **2.987** | **3.834** | **2.723** | **3.750** | **3.364** | **3.102** | 43 | 2848 |
| Llava-Next-Video-7B | Vanilla | 3.217 | 2.833 | 3.484 | 2.136 | 2.938 | 2.922 | 2.518 | 27 | 4608 |
| | SlowFast | 2.985 | 2.613 | 3.234 | 2.441 | 3.082 | 2.871 | 2.375 | 30 | 3680 |
| | Ours | **3.399** | **3.030** | **3.645** | **2.459** | **3.419** | **3.190** | **2.719** | 34 | 2848 |
| llava-v1.6-vicuna-13b | Vanilla | 2.977 | 2.6323 | 3.2635 | 2.1844 | 3.3826 | 2.88796 | 2.5509 | 38 | 4608 |
| | SlowFast | 2.9391 | 2.6862 | 3.1414 | 2.188 | 3.3507 | 2.8610 | 2.6034 | 37 | 3680 |
| | Ours | **3.1653** | **2.8126** | **3.4539** | **2.1883** | **3.4429** | **3.0126** | **2.6132** | 42 | 2848 |
| Llava-Next-Interleave-7B-DPO | Vanilla | 3.876 | 3.203 | 4.044 | 3.160 | 3.792 | 3.729 | 3.231 | 19 | 6272 |
| | SlowFast | 3.643 | 2.929 | 3.827 | 2.920 | 3.711 | 3.406 | 3.212 | 22 | 6230 |
| | Ours | **3.771** | **3.152** | **3.933** | **3.008** | **3.719** | **3.517** | **3.224** | 23 | **4134** |
| Llava-onevision-qwen2-0.5b-ov | Vanilla | 3.147 | 2.709 | 3.399 | 2.411 | 2.944 | 2.922 | 2.764 | 40 | 6272 |
| | SlowFast | 2.911 | 2.607 | 3.223 | 2.248 | 2.828 | 2.763 | 2.533 | 36 | 6230 |
| | Ours | **3.118** | **2.708** | **3.391** | **2.473** | **3.174** | **2.973** | **2.796** | 49 | **4134** |
| llava-onevision-qwen2-7b | Vanilla | 3.5942 | 3.1708 | 3.8191 | 2.8637 | 3.5210 | 3.3937 | 3.2305 | 39 | 6272 |
| | SlowFast | 3.5506 | 3.1232 | 3.7445 | 2.6152 | 3.3647 | 3.2796 | 3.2124 | 43 | 6230 |
| | Ours | **3.9709** | **3.3953** | **4.1042** | **3.2244** | **3.7555** | **3.6900** | **3.2244** | 45 | **4134** |

response, are crucial. Furthermore, relying on a static video representation fails to accommodate the LLM's dynamically changing focus during generation. Thus, adaptively adjusting the LLM's focus during inference, as implemented in our Adaptive Token Adjustment, is necessary to balance generation accuracy and efficiency.

**Comparison of the Generated Answer Between ZoomVLM and Vanilla VLM.** To provide a qualitative comparison between ZoomVLM and the baseline vanilla video VLM, we visualize a sample generated by both ZoomVLM and the vanilla Llava-Next-Video-7B-DPO model, as shown in Fig. 3 (additional visualizations are available in App. I). We observe that the responses generated by ZoomVLM contain more detailed information (highlighted in **bold**) compared to those from the vanilla model. We attribute this enhancement to ZoomVLM's ability to selectively include a small set of high-resolution frames and to adaptively adjust tokens to better focus on areas of interest during generation. Consequently, even though ZoomVLM uses fewer tokens, it provides more detailed responses by emphasizing critical information, enabling the video VLM to focus on essential content. This facilitates more accurate and detailed outputs.

**Visualization on the Video Overview Augmenter Frame Selection Strategy.** By analyzing the highlighted orange frames selected by the Video Overview Augmenter, we observe that it effectively selects important frames while avoiding interference. Specifically, we note the following

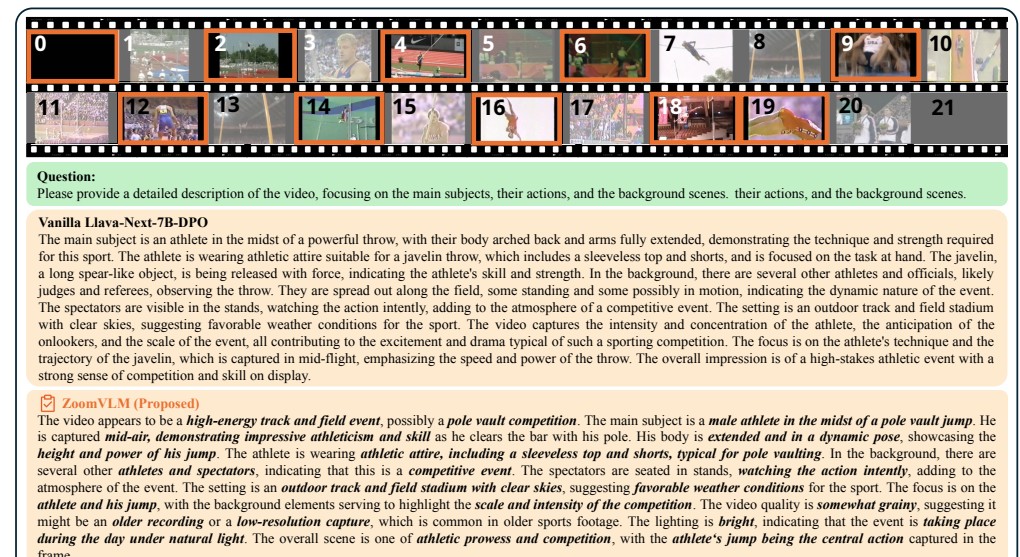

Figure 3: Comparison of content generated by the vanilla Llava-Next-7B-DPO model (Li et al., 2024a) and the same model enhanced with our proposed ZoomVLM framework. The video frames highlighted with orange bounding box are frames selected by the Video Overview Augmenter to augment the video overview.

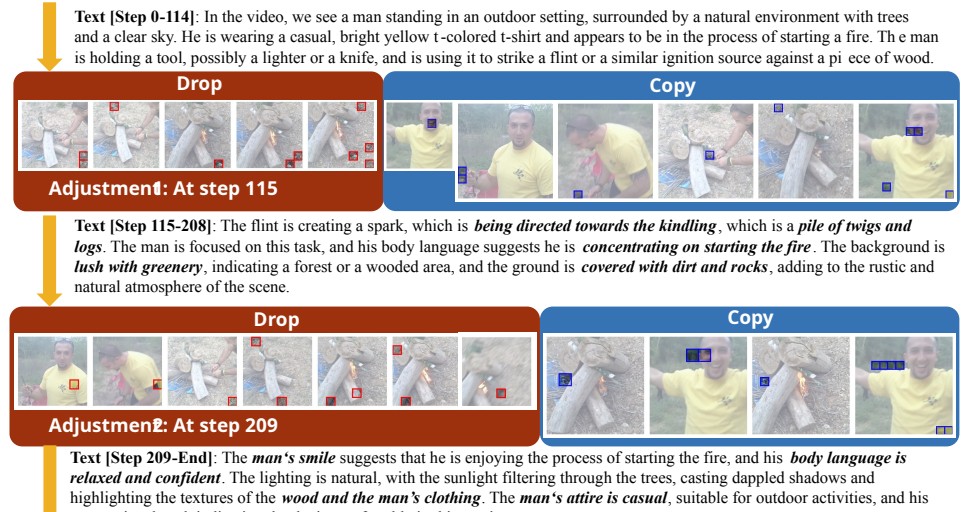

Figure 4: Visualization of the Adaptive Token Adjustment strategy throughout the generation process. At each adjustment step, red bounding boxes indicate frame regions corresponding to dropped tokens, while blue bounding boxes highlight regions with copied tokens. High-resolution frames are displayed in a larger size, and text closely related to the copied tokens is highlighted in **bold**.

observations: (1) Omission of Non-Unique Important Frames: Frames that seem important but lack unique information are omitted. For example, the 3rd frame features a prominent human figure but is not selected because the model avoids content already covered. The absence of distinctive activity makes this frame unnecessary. (2) Exclusion of Irrelevant Detail: Frames overloaded with irrelevant details are excluded. The 11th frame is disregarded because the video's background and environment have been sufficiently summarized, adding no new information. (3) Even Temporal Distribution: The selected frames are evenly spaced over time, aligning with typical human expectations for important content distribution. This minimizes redundancy; for instance, distinct frames like the 7th and 16th are not both chosen, demonstrating the system's efficiency across the video timeline.

**Visualization on Adaptive Token Adjustment Strategy.** We further visualize the Adaptive Token Adjustment strategy during the generation process, as shown in Fig. 4. By leveraging attention mechanisms as identifiers, ZoomVLM effectively discerns unnecessary tokens to drop and important tokens to retain. Specifically, after each adjustment operation, the subsequent generated content focuses more on important, detailed information at a relatively small scale, such as "being directed towards the kindling". Interestingly, contrary to the common understanding that low-resolution frames have less redundancy than high-resolution frames, the Adaptive Token Adjustment module tends to aggressively drop tokens in low-resolution frames while copying tokens in high-resolution frames. We hypothesize that this is because the Video Overview Augmenter accurately selects high-resolution frames to augment, thus providing a better, more detailed representation.

## 5.3 Ablation Studies

**Ablation and Visualization on the Frame-Wise Attention Distribution Across Different Layers.** Fig. 5 (a), shows the averaged frame-wise attention distribution across layers from all

Table 2: Ablate on the selection of **L**.

| Layer Range | Vanilla | [1-15] | [5-20] (**Ours**) | [15-30] |
|---|---|---|---|---|
| VDD Score | 2.843 | 3.082 | 3.102 | 3.084 |

VDD dataset samples. We observe that early and deep layers concentrate their attention on later frames, while middle layers distribute attention more uniformly across frames, which is better aligned with the human perception process. To validate whether this helps with identifying important frames, we visualize the top-8 high-attention frames selected by an early layer (4th), a middle layer (15th), and a deep layer (25th) in Fig. 5 (b). We observe that the frames selected by the middle layer have a better diversity compared with other layers. Then, we conduct an ablation study using **L** consisting of different layers and validate their impact on the performance of ZoomVLM, as shown in Table 2, aligned with our observation and analysis above, middle layers help with the achieved score while other layer ranges suffer from drop in score on VDD due to the lack of accurate indicator to find what is the critical frames that need to be leveraged to augment the video overview.

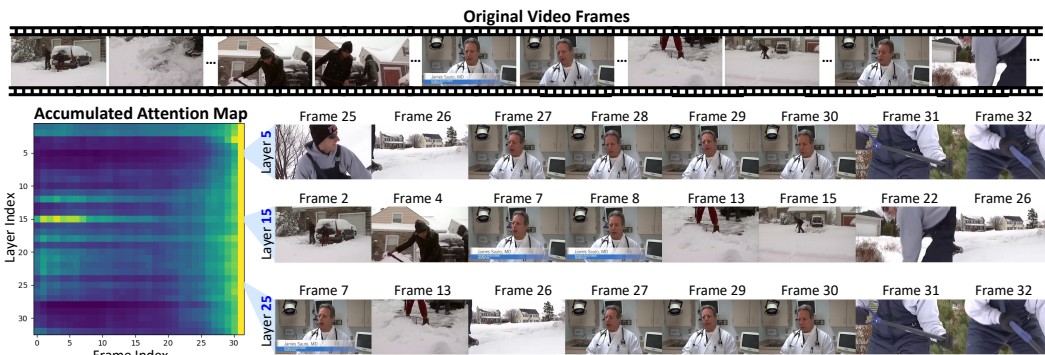

Figure 5: Visualization on attention distribution and layer-wise high-attention frames.

**Performance Breakdown.** ZoomVLM framework integrates two key components, to study each component's contribution to the final performance, we conduct an ablation study to validate the performance breakdown across different components. As shown in Table 3,

Table 3: Performance breakdown of ZoomVLM.

| Setting | Vanilla | Summary Only | W/o Adjustment | ZoomVLM |
|---|---|---|---|---|
| VDD Score | 2.843 | 2.801 | 3.044 | 3.102 |

Table 4: Ablate on the selection of $k$.

| # Frames | 5 | 8 | 10 (Ours) | 12 |
|---|---|---|---|---|
| VDD Score | 3.026 | 3.048 | 3.102 | 3.084 |

**Ablation on the Achieved Accuracy-Efficiency Trade-Off of ZoomVLM.** Inserting more high-attention frames (i.e., larger $k$) can potentially provide more detailed information for the VLM to better understand the video at the cost of a higher inference cost. To explore a better accuracy-efficiency trade-off, we conduct an ablation study on the selection of $k$. As shown in Table 4, the accuracy improvement saturates when $k > 10$ and thus, we adopt $k = 10$ as our default choice.

**Ablation on Effectiveness of Important Frame Identification in Video Overview Augmenter.** To validate the effectiveness of our approach in identifying important frames, we conduct experiments with different frame selection techniques, including the commonly used uniform selection Xu et al. (2024b) and random selection. As shown in Table 5, although the idea of augmenting the video with high-resolution information is generally helpful, our method achieves a 0.100∼0.116 higher VDD score compared to different baselines.

**Ablation on the Copy and Drop Interval.** Although Adaptive Token Adjustment is beneficial, practical constraints prevent its implementation at every step. To determine an effective interval for this adjustment, we conducted an ablation study using the Llava-Next-Video-7B-DPO model at VDD. The results, presented in Table 6, indicate that while adjustments generally enhance the BDD score, a fixed interval of implementation degrades performance compared to our strategy of adjusting every three sentences. We hypothesize that a predetermined interval may disrupt the KV cache during sentence generation, potentially leading to consistency issues.

Table 5: Ablate on approaches to selection important frames to augment.

| Approach | Uniform | Random | Ours |
|---|---|---|---|
| VDD Score | 3.002 | 2.986 | 3.102 |

**More ablations can be found in App. F.**

Table 6: Ablate on the copy and drop interval.

| Interval | Baseline | 30 | 50 | 70 | Ours |
|---|---|---|---|---|---|
| VDD Score | 2.843 | 2.972 | 2.929 | 2.959 | 3.102 |

## 6 CONCLUSION

In this paper, we introduced ZoomVLM, a tuning-free, plug-and-play efficient video processing framework for video VLMs. Inspired by human perceptual strategies, ZoomVLM first generates a high-level overview of the entire video and then adaptively zooms in on specific parts based on the content being generated. Our framework incorporates two key components: a Video Overview Augmenter, which creates an informative summary by augmenting downsampled videos with high-resolution keyframes, and an Adaptive Token Adjustment mechanism, which predicts the importance of different video segments and adjusts token allocation accordingly during inference. Extensive experiments demonstrate that ZoomVLM improves accuracy efficiency trade-offs by as much as 30% higher token generation rate a 0.259 improvement in the Video Detail Description score.

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

# A MEMORY OVERHEAD OF ZOOMVLM

To provide a more comprehensive evaluation of the achieved efficiency of ZoomVLM, we further benchmark the peak memory consumption of ZoomVLM with other baseline solutions. As shown in Table 7, ZoomVLM reduces peak memory usage by 29% to 32% compared to the most competitive baseline (i.e., Slowfast-llava Feichtenhofer et al. (2019)). This set of experiments further highlights the comprehensive efficiency improvements of ZoomVLM, covering both latency and memory overhead.

Table 7: Benchmarking memory overhead of ZoomVLM and baseline solutions.

| Model | Method | Peak Memory |
|---|---|---|
| **Llava-Next-Video-7B-DPO** | vanilla | 62.89GB |
| | slow-fast | 36.46GB |
| | ours | 25.04GB |
| **llava-v1.6-vicuna-13b** | vanilla | OOM |
| | slow-fast | 60.4GB |
| | ours | 42.95GB |

# B ZOOMVLM LATENCY PROFILING

To better illustrate ZoomVLM's efficiency, we profiled the latency of each module using the Llava-Next-Video-7B-DPO model on the VDD dataset. As shown in Table 8, for an output length of 300 tokens, the Video Overview Augmenter and Adaptive Token Adjustment modules account for less than 5% and 1% of the total inference cost, respectively. Despite this, the improved token efficiency introduced by these two modules leads to a $\sim$25% reduction in the total inference cost, sourced from $\sim$80% less latency in the prefilling stage and a $\sim$20% less latency in the auto-regressive generation stage, thanks to a $\sim$40% fewer tokens needed to represent video.

Table 8: Profile ZoomVLM and vanilla Llava-Next-Video-7B-DPO on VDD dataset.

| Model | Video Overview Augmenter | Adaptive Token Adjustment | Backbone (Autoregressive) | Backbone (Prefill) | Total time | VDD Score |
|---|---|---|---|---|---|---|
| **Vanilla** | 0 | 0 | 7.833 | 1.081 | 8.915 | 2.843 |
| **ZoomVLM** | 0.327 | 0.070 | 6.279 | 0.221 | 6.898 | 3.102 |

# C EVALUATION OF ZOOMVLM ON ADDITIONAL BENCHMARKS

To further validate the general capability of ZoomVLM across different tasks and evaluation settings, we further benchmark ZoomVLM on MLVU Zhou et al. (2024) and AuroraCap Chai et al. (2024). As shown in the tables below, ZoomVLM consistently achieves comparable or superior accuracy while demonstrating improved efficiency compared to the baseline solution. These results further validate the generalizability of ZoomVLM across diverse benchmarks.

Table 9: Benchmark ZoomVLM on AuroraCap dataset.

| Model | method | background | camera | detailed | main_object | short | Token/sec |
|---|---|---|---|---|---|---|---|
| Llava-Next-Video-7B-DPO | Vanilla | 38.55 / 2.0008 | 37.68 / 1.951 | 42.91 / 2.2238 | 40.88 / 2.0954 | 41.63 / 2.1500 | 27 |
| | Ours | 38.5 / 1.9905 | 37.6 / 1.9353 | 42.5 / 2.2036 | 40.97 / 2.1136 | 41.6 / 2.1486 | 32 |

Table 10: Benchmark ZoomVLM on long video understanding MLVU dataset.

| Model | Method | SSC | VS | G-Avg | Token/sec |
|---|---|---|---|---|---|
| Llava-Next-Video-7B-DPO | Vanilla | 3.5743 | 2.6523 | 3.1132 | 27 |
| | Ours | 3.5095 | 2.6714 | 3.09045 | 31 |

## D DETAILED EVALUATION SETTINGS

**Video Detailed Description** We evaluate VLMs on Video Detailed Description via the following settings: 22 low-resolution frames, 10 high-resolution frames, adding one newline token after all frames, 2 for pooling stride, pooling before projection, and 1024 for max generation tokens.

**Video-ChatGPT** We use the same settings as Video Detailed Description for VideoGPT: 22 low-resolution frames, 10 high-resolution frames, adding one newline token after all frames, 2 for pooling stride, pooling before projection, and 1024 for max generation tokens.

**Discussion on Scoring GPT Model** Because the default evaluator gpt-3.5-turbo-0613 has been deprecated, we use gpt-3.5-turbo to score all the results of Video Detailed Description and Video-GPT.

## E PSEUDOCODE FOR ZOOMVLM

---
**Algorithm 1** Pseudocode for ZoomVLM
---
**Require:** Input video $\mathcal{V}$, initial generation step $s$, pretrained VLM consisting of ViT $\mathcal{M}_V$, projection layer $\mathcal{M}_P$, and LLM $\mathcal{M}_L$
  {V}ideo Overview Augmenter
  Generate a compact video overview $\hat{\mathbf{V}}$ with downsampling
  Generate the token representation $\hat{\mathbf{P}}$ by passing $\hat{\mathbf{V}}$ to $\mathcal{M}_V$ and $\mathcal{M}_P$, sequentially
  Pass $\hat{\mathbf{P}}$ to $\mathcal{M}_L$ to generate $s$ tokens
  Identify critical frames $\mathcal{A}$ following Eq. 8, obtain their token representation $\mathbf{P}_C$
  Concatenate $\hat{\mathbf{P}}$ and $\mathbf{P}_C$ to resume the generation process, as in Eq. 9, where the token representation of $\mathbf{T}$, $\hat{\mathbf{P}}$, $\mathbf{P}_C$ can be reused, but previously generated $s$ tokens need to be regenerated {S}tart autoregressive generation
  **for** $t \leftarrow 1$ to $T$ **do**
    **if** not end of sentence **then**
      Generate $x_t$
    **else**
      {S}tart Adaptive Token Adjustment after finishing generating each sentence
      Check the attention map of the current state to identify high-attention and low-attention tokens following Eq. 11
      Adjust token in KV cache following Eq. 12
      Continue generation with updated tokens in KV cache following Eq. 13, in this step, all remaining KV cache will inherit the previously generated one, no recalculation is needed
      Generate $x_t$
    **end if**
  **end for**
  **return** $\{x_0, \cdots, x_{T-1}\}$
---

## F ADDITIONAL ABLATION STUDIES

### F.1 ABLATION ON NUMBER OF COPY AND DROPPING TOKENS IN EACH ADJUSEMENT

Copying and dropping a larger number of tokens during each Adaptive Token Adjustment allows for a more dynamic inference process and more aggressive manipulation of the LLM's attention. To investigate this, we conduct an ablation study to evaluate the impact of the number of tokens copied and dropped in each adjustment. As shown in Table 11, we observe that increasing the number of tokens copied and dropped does not necessarily lead to better performance. We attribute the inferior performance to excessive changes in the token distribution, which may confuse the VLM and lead to hallucinations. Furthermore, the superior accuracy achieved when fewer tokens are changed suggests that, despite the thousands of tokens in the video representation, only a few are truly important. Therefore, a relatively small manipulation of the tokens can significantly enhance the achievable accuracy.

Table 11: Ablate the number of tokens to copy and drop in each adjustment.

| # of Tokens | 10 (Ours) | 20 | 30 |
|---|---|---|---|
| VDD Score | 3.102 | 3.036 | 3.062 |

## F.2 ABLATION ON THE SELECTION OF THE NUMBER OF INITIAL GENERATION STEP $s$

An important hyperparameter in the Video Overview Augmenter is $s$, which controls the number of generation steps that must be performed before the VLM acquires sufficient knowledge about the frame-wise importance of the video for subsequent generation steps. To validate our selection of $s$, we conduct an ablation study using different values of $s$, as shown in Table 12, Consistent with our analysis in Section 4.2, we find that deciding the importance of frames too early (i.e., smaller $s$) leads to a drop in accuracy due to insufficient understanding of the video and inadequate planning for future content generation. When $s$ exceeds 15 steps, the quality of the selected frames stabilizes. Therefore, we select $s = 20$ in ZoomVLM to provide a margin for scenarios that are difficult to understand and require careful planning.

Table 12: Ablate the selection of $s$.

| $s$ | 5 | 10 | 15 | 20 (Ours) | 25 |
|---|---|---|---|---|---|
| VDD Score | 2.886 | 2.970 | 3.056 | 3.102 | 3.038 |

## F.3 VALIDATION OF THE VIDEO OVERVIEW FORMAT

We further examined the efficacy of our chosen video overview format, which integrates a high-level overview with a few keyframes. As shown in Table 13, our approach outperforms commonly used methods such as spatial pooling only and temporal sampling only in terms of VDD score, while maintaining comparable token efficiency. We have added this experiment to the appendix of our paper.

Table 13: Comparison of settings with different video processing approaches.

| Setting | Original Video | Spatial Pooling | Temporal Sampling | Video Overview Augmenter |
|---|---|---|---|---|
| # Tokens | 4608 | 2048 | 2880 | 2848 |
| VDD Score | 2.843 | 2.346 | 2.727 | 2.801 |

## F.4 ABLATION ON REDUCING TOKENS WITH ONLY ADAPTIVE TOKEN ADJUSTMENT

We conducted additional experiments using only the Adaptive Token Adjustment module to reduce the number of video tokens with varying reduction rates. As shown in the table below, slight adjustments (e.g., dropping and copying fewer than 30 tokens per adjustment) improved the VDD score (e.g., from 0.006 to 0.032). However, more aggressive adjustments led to a significant performance drop (e.g., from 0.102 to 0.909), confirming that extreme token reductions negatively impact model performance. Thus, it is critical to first leverage the Video Overview Augmenter to generate a video overview and largely reduce the number of video tokens, then introduce the Adaptive Token Adjustment to calibrate video representation and further improve the response accuracy.

## G EXPERIMENT SETTINGS FOR FIG. 1

In this experiment, we adhere to the common practices established by SOTA video VLMs as detailed in recent literature (Li et al., 2024a; Xu et al., 2024b; Zhang et al., 2024b; Li et al., 2024c). Our methodology involves three types of downsampling, spatial, temporal, and hybrid. Specifically:

(1) For spatial downsampling, we begin with videos at a resolution of 336x336 pixels, using a patch size of 14 and a stride of 2, as specified in Li et al. (2024a). We systematically reduce the resolution to 224 and subsequently to 168, while maintaining all other parameters constant.

Table 14: Ablation on Reducing Tokens with Only Adaptive Token Adjustment.

| Total # of Reduced Tokens | 0 | 15 | 30 | 75 | 105 | 150 | 300 |
|---|---|---|---|---|---|---|---|
| # of Dropped Tokens | 0 | 10 | 20 | 50 | 70 | 100 | 200 |
| # of Copied Tokens | 0 | 5 | 10 | 25 | 35 | 50 | 100 |
| VDD Score | 2.843 | 2.842 | 2.8 | 2.741 | 2.386 | 2.148 | 1.933 |

(2) In temporal downsampling, we start with the default setting of 32 frames, following the guidelines in Li et al. (2024a), and progressively decrease the frame count to 24, 16, and finally 8, without altering any additional parameters.

(3) For hybrid downsampling, consistent with the pioneering approach in Xu et al. (2024b), we construct two distinct processing branches. In the high-resolution branch, each frame, initially at 336x336 resolution, undergoes a 1x2 pooling, reducing each frame to a 24x12 token grid. In the low-resolution branch, starting at the same initial resolution, each frame is subjected to a 6x6 pooling, resulting in a 4x4 token grid. Our experimental setup evaluates three configurations: the first with 10 frames in the high-resolution branch and 50 frames in the low-resolution branch, resulting in a token representation of 3680 tokens; the second with 5 frames in the high-resolution branch and 45 frames in the low-resolution branch, resulting in a token representation of 2160 tokens; and the third with 3 frames in the high-resolution branch and 25 frames in the low-resolution branch, resulting in a token representation of 1264 tokens.

## H ADDITIONAL VISUALIZATION ON ATTENTION DISTRIBUTION

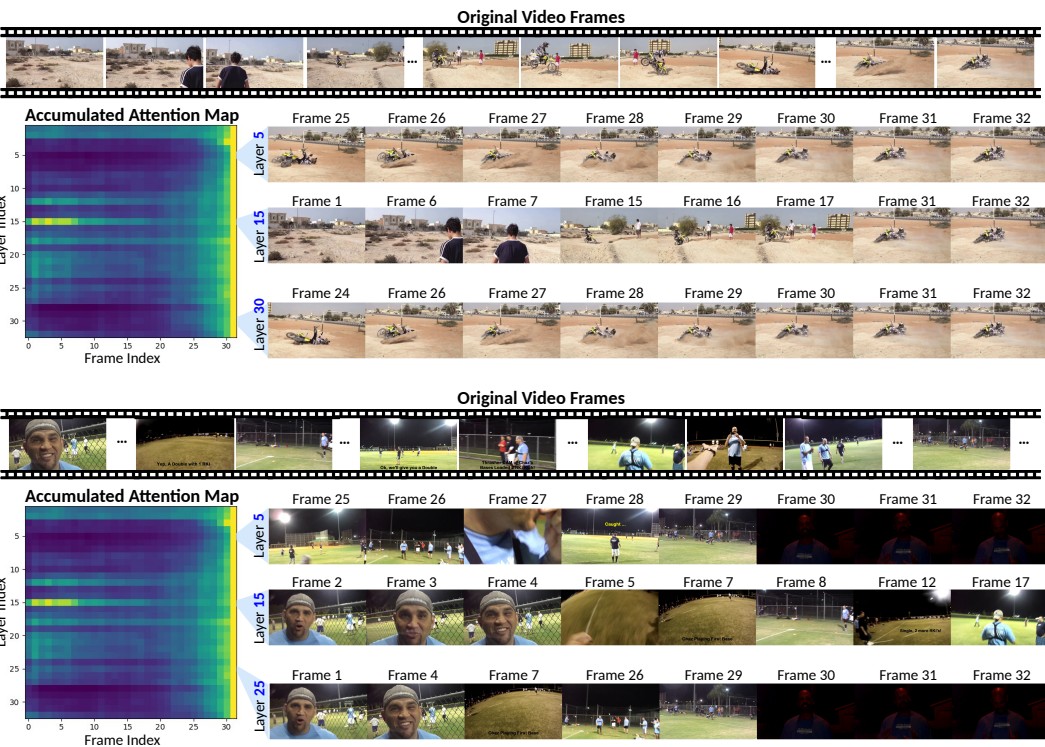

Figure 6: Additional visualizations on the attention distribution and layer-wise high-attention frames.

## I ADDITIONAL VISUALIZATION ON THE GENERATED CONTENT

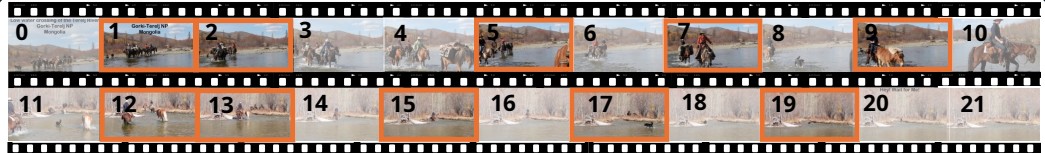

**Question:**
Please provide a detailed description of the video, focusing on the main subjects, their actions, and the background scenes. their actions, and the background scenes.

**Vanilla Llava-Next-7B-DPO**
A DIY project, where they are working on a task that involves cutting or shaping materials with precision. The main subjects are individuals who are focused on their tasks, with their hands and the materials they are working on in the foreground. The background is relatively nondescript and does not provide much detail, but it seems to be an indoor setting with a table or work surface where the activity is taking place. The individuals are wearing casual clothing, and one person is wearing a camouflage pattern, suggesting a relaxed or informal setting. They are using tools that are not clearly visible but could be scissors or other cutting instruments, and they are handling materials that look like paper or thin sheets. The atmosphere is one of concentration and teamwork, with people working together to achieve a common goal. The lighting is bright and even, indicating an indoor environment with ample lighting, which is conducive to the task at hand. The video does not provide any audio, so the focus is entirely on the visuals, and the actions of the individuals are the primary focus. The camera angle is stable and stationary, capturing the hands and the materials in detail, allowing for a clear view of the process. The overall impression is of a collaborative effort, possibly in a creative or educational setting, where the participants are working on a hands-on task that requires precision and attention to detail.

**ZoomVLM (Proposed)**
The video depicts a group of individuals engaged in a collaborative activity, possibly a nail painting session or a similar social event. ***The main subjects are individuals whose hands are visible in the frame, with one person actively painting another's nails.*** The person receiving the nail painting is seated and appears to be relaxed, with their hands open and ready for the process. ***The person doing the painting is focused on their task, holding a small object, which could be a nail file or a nail polish brush, and is applying it to the nails of the other person.*** The background is relatively nondescript, with a plain surface that could be a table or a countertop, and there are other individuals partially visible in the frame, suggesting a casual, communal setting. ***The lighting is soft and even, and the overall atmosphere appears to be calm and relaxed.*** There are no distinctive background elements that provide additional context or setting details, which keeps the focus on the interaction between the individuals and their activity. The video captures a moment of personal care and connection between the participants, emphasizing the shared experience of the activity.

**Question:**
Please provide a detailed description of the video, focusing on the main subjects, their actions, and the background scenes. their actions, and the background scenes.

**Vanilla Llava-Next-7B-DPO**
Natural landscape. The riders are dressed in casual attire suitable for outdoor activities, with some wearing hats and jackets, suggesting a cool or possibly mountainous climate. The horses are in motion, walking through the water, which is shallow and calm, indicating a gentle current. The riders maintain a relaxed posture, suggesting a leisurely pace rather than a rushed journey. The background is lush with trees and foliage, indicating a forested area, and the sky is clear with a few scattered clouds, suggesting a sunny day. The overall atmosphere is serene and peaceful, with the sound of the water and the natural environment adding to the tranquility of the scene. The riders appear to be enjoying their time, possibly engaging in a recreational activity or exploring the area on horseback.

**ZoomVLM (Proposed)**
***The video captures a serene scene of a group of people riding horses across a shallow river.*** The riders are dressed in traditional attire, suggesting a sense of adventure or perhaps a cultural event. ***The horses are spread out across the width of the river, some closer to the camera and others further away, creating a sense of depth in the image.*** The riders appear to be in motion, guiding their horses through the water, which adds a dynamic element to the scene. The river itself is calm, reflecting the tranquility of the environment. The background is lush with trees and foliage, indicating a natural, possibly rural setting. The trees are tall and lush, suggesting a healthy ecosystem. The riverbanks are lined with vegetation, adding to the natural beauty of the scene. The sky is clear and blue, suggesting good weather conditions for the ride. The overall atmosphere is peaceful and idyllic, with the sound of the water and the gentle movement of the horses adding to the calming ambiance. The riders are spread out, some closer to the camera and others further away, which gives a sense of depth to the image. The riders' posture suggest they are experienced and comfortable with their horses, guiding them through the water with ease. ***The horses are walking through the water, their bodies partially submerged, indicating a shallow river.*** The riders' attire and the horses' calm demeanor suggest a harmonious relationship between the humans and the animals. The video captures a moment of connection between humans and nature, showcasing the beauty of the outdoors and the bond between humans and their animals. The riders' attire and the horses' calm demeanor suggest a sense of respect and trust between the two. The overall scene is one of tranquility and harmony, with the riders and horses moving in unison through the water, creating a picturesque and peaceful moment.

Figure 7: Additional visualizations on the comparison of the content generated by the vanilla Llava-Next-7B-DPO model Li et al. (2024a) and same model enhanced with our proposed ZoomVLM framework.

