# OpenReview forum: "ZoomVLM: A Tuning-Free Framework for Efficient Video Understanding via Adaptive Zooming in Vision-Language Models"
_ICLR.cc/2025/Conference — Submitted to ICLR 2025_

### Official Review · Reviewer_QnQy · 2024-10-28

**Soundness:** 1
**Presentation:** 2
**Contribution:** 2
**Rating:** 5
**Confidence:** 5

**Summary:**

This paper proposes ZoomVLM, a tuning-free, plug-and-play efficient video processing framework for video VLMs. ZoomVLM integrates two key components: (1) a Video Overview Augmenter, which enables cost-effective high-level understanding by augmenting downsampled video overview with a few high-resolution keyframes; and (2) an Adaptive Token Adjustment, which predicts the importance of different video parts in the upcoming generation process and adjusts the number of tokens allocated to each part according to their importance.

The contributions are summarized as follows:
1. propose a tuning-free, plug-and-play efficient video processing pipeline for VLMs, dubbed ZoomVLM.
2. ZoomVLM integrates two key components to efficiently select necessary information by leveraging the attention distribution within the VLM: Video Overview Augmenter and Adaptive Token Adjustment.
3. Extensive experiments demonstrate significant improvements.

**Strengths:**

1. This paper observes a bottleneck in the practical application of current video VLMs, namely the excessive number of visual tokens, which severely affects inference speed. The paper attempts to address this issue with a tune-free approach, which is a good starting point.
2. The ablation experiments are quite comprehensive.

**Weaknesses:**

1. There should be a section that analyzes the efficiency of the method proposed in the paper, including how many tokens have been reduced, which KV caches can be reused or recalculated, the theoretical speedup ratio, etc. It would be best to include pseudo code for inference.
2. The results for Vanilla in Table 1 are significantly lower than the official results for the LLaVA-NeXT-Video series, and the authors do not explain the reason in the paper. Is it due to resolution, retraining, or some other factor, and the paper's results also fail to surpass the official results for the LLaVA-NeXT-Video series. This makes it difficult to believe in the effectiveness of the methods presented in the paper.
3. The paper only includes two benchmarks and llava series models, and more benchmarks and models could be added to enhance the credibility and trustworthiness of the method proposed, as perceived by the readers.

**Questions:**

1. The paper mentions resuming the generation process, but video tokens will undergo corresponding changes, such as concatenating $P_C$. In this scenario, can KV caches still be reused, and will the computational load increase?
2. In Equation 12, duplicating tokens does not introduce additional information; why would it be effective?

---

> ### Author Response · Authors · 2024-11-25
> **Response to Reviewer QnQy**
>
> We greatly appreciate your review efforts. Thank you for your encouraging recognition of ZoomVLM's identified efficiency bottleneck and acknowledgement of our comprehensive ablation study! Below, we address your questions/comments and provide detailed clarifications:
>
> **W1**: Efficiency Analysis and Inclusion of Pseudocode
>
> **A1**: Thank you for the suggestion! We have expanded the analysis of ZoomVLM's efficiency and addressed the points you raised as follows:
>
> > (1) **Token reduction**: The token reduction is achieved by the Video Overview Augmenter. As shown in Table 1 of our manuscript, ZoomVLM **reduces 1760\~2138 video tokens, accounting for 34%\~39% of total video tokens**.
>
> > (2) **KV cache reuse**:
>
> >> (a) During the **Video Overview Augmenter**, the KV cache of the condensed video overview ($\hat{P}$) can be preserved, while only the short sequence ($s$ tokens) generated based on $\hat{P}$  needs to be recalculated.
>
> >> (b) During **Adaptive Token Adjustment**, all KV caches can be reused, further enhancing efficiency.
>
>
> > (3) **Theoretical Speedup**: The generation process of VLMs is significantly constrained by data movement due to the large KV cache. The theoretical speedup of ZoomVLM correlates with the reduction in KV cache size, which corresponds to a **34%–39% improvement** compared to the baseline.
>
> > (4) **Memory overhead reduction**: ZoomVLM also reduces memory overhead significantly. When generating 300 tokens, it achieves **a 29%–32% reduction in peak memory usage compared to the most competitive baseline (i.e., SlowFast-Llava)**. These results further demonstrate that ZoomVLM improves both latency and memory efficiency. We have included this experiment in the appendix of our manuscript.
> | Models                  | Method      | Peak Memory (300 tokens) |
> |-------------------------|-------------|---------------------------|
> | Llava-Next-Video-7B-DPO | Vanilla     | 62.89GB                  |
> |                         | Slow-Fast   | 36.46GB                  |
> |                         | Ours        | 25.04GB                  |
> | llava-v1.6-vicuna-13b   | Vanilla     | OOM                      |
> |                         | Slow-Fast   | 60.4GB                   |
> |                         | Ours        | 42.95GB                  |
>
> > (5) **Latency Profiling**: To better illustrate ZoomVLM's efficiency, we profiled the latency of each module using the Llava-Next-Video-7B-DPO model on the VDD dataset. For an output length of 300 tokens, **the Video Overview Augmenter and Adaptive Token Adjustment modules account for less than 5% and 1% of the total inference cost, respectively**. Despite this, the improved token efficiency introduced by these two modules leads to a **~25% reduction in the total inference cost**, sourced from ~80% less latency in the prefilling stage and a ~20% less latency in the auto-regressive generation stage, thanks to a ~40% fewer tokens needed to represent video. We have included these profiling results in the appendix of our manuscript.
> | Method   | Video Overview Augmenter (s) | Adaptive Token Adjustment (s) | Backbone Inference (Autoregressive) | Backbone Inference (Prefill) | Total Inference Time | VDD Score |
> |----------|------------------------------|--------------------------------|--------------------------------------|-------------------------------|----------------------|-----------|
> | Vanilla  | 0                            | 0                              | 5.24                                 | 1.0812                       | 5.24                 | 2.843     |
> | ZoomVLM  | 0.3277                       | 0.38                           | 4.3                                  | 0.2215                       | 5.0077               | 3.102     |
>
> > (6) **Pseudocode**: To facilitate a clearer understanding of ZoomVLM, we have included pseudocode for its inference process in the appendix of our paper, as per your suggestion. You can also access the pseudocode here: https://imgur.com/d3WTJxM

---

> ### Author Response · Authors · 2024-11-25
>
> **W2**: Discrepancy in Baseline Results and Comparison with Official LLaVA Series
>
> **A2**: Thank you for raising this important point! The discrepancy in baseline evaluation results arises from a change in the judge model. The official LLaVA series used **GPT-3.5-turbo-0613** as the judge model, but this version has been recently retired [1]. For our evaluations, we switched to **GPT-3.5-turbo**, which led to differences in absolute performance scores.
> To address this concern, we conducted additional experiments using various GPT variants to verify that the relative trends between ZoomVLM and the baseline solutions remain consistent. The results, summarized in the table below, confirm that despite variations in absolute scores across different GPT versions, ZoomVLM consistently outperforms the vanilla baseline, demonstrating the robustness of our approach.
>
> Additionally, to provide transparency and facilitate further exploration, we have released a code sample to evaluate the original LLaVA model with different GPT variants. This resource allows for a deeper understanding of the impact of judge model variations on evaluation results. You can access the code here: https://anonymous.4open.science/r/Efficient_VLM-5C88/
>
> | Model                     | LLM-evaluator | Method   | Correctness | Detail  | Context | Temporal | Consistency | Average  |
> |---------------------------|---------------|----------|-------------|---------|---------|----------|-------------|----------|
> | Llava-Next-Video-7B-DPO   | gpt-4o        | Vanilla  | 3.4596      | 2.8808  | 3.7705  | 2.6974   | 3.2031      | 3.2023  |
> |                           |               | ZoomVLM  | 3.5726      | 2.9684  | 3.8632  | 2.7761   | 3.8475      | 3.4056  |
> |                           | gpt-4o-mini   | Vanilla  | 3.4374      | 2.9153  | 3.773   | 2.6994   | 3.1992      | 3.2049  |
> |                           |               | ZoomVLM  | 3.5576      | 2.9760   | 3.8547  | 2.7896   | 3.7555      | 3.3866  |
> |                           | gpt-3.5-turbo | Vanilla  | 3.0937      | 2.6007  | 3.511   | 2.3126   | 3.1864      | 2.9409  |
> |                           |               | ZoomVLM  | 3.5286      | 2.9865  | 3.8337  | 2.7234   | 3.7495      | 3.3643  |
>
>
>
> **W3**: Evaluation of more benchmarks and models
>
> **A3**: Thank you for this valuable suggestion! Regarding the evaluation of ZoomVLM on the LLaVA series models, we would like to clarify that the LLaVA series represents the state-of-the-art (SOTA) in video VLM frameworks. Our focus is on improving the efficiency of SOTA models, making the LLaVA series a natural choice for evaluation. This approach aligns with the common practice in video VLM-related research [2, 3, 4].
>
> To validate the effectiveness of ZoomVLM, we evaluated it across multiple versions of the LLaVA model, which vary significantly in their training pipelines and capabilities. Additionally, we tested ZoomVLM on models with different backbone LLMs, providing a comprehensive evaluation of ZoomVLM across a diverse range of settings.
>
> Following your suggestion, we expanded our experiments to include additional video VLMs, such as LLaVA-OneVision-Qwen2-7B and LLaVA-v1.6-Vicuna-13B, and evaluated ZoomVLM on more benchmarks, including MLVU [5], which focuses on long video understanding tasks, and AuroraCap [6]. As shown in the table below, ZoomVLM consistently achieves comparable or superior accuracy while demonstrating improved efficiency compared to baseline solutions. We have included these experiments in the appendix of our manuscript.
>
> | Setting                      | Method   | SSC     | VS      | G-Avg   | Token/Sec   | Peak Memory Overhead   |
> |-----------------------------|----------|---------|---------|---------|---------|---------|
> | Llava-Next-Video-7B-DPO@MLVU     | Vanilla  | 3.5743  | 2.6523  | 3.1133  |  25    | 62.89 GB   |
> |                             | Ours     | 3.5095  | 2.6714  | 3.09045 |  32  |   25.04GB   |
>
>
> | Model                       | Method   | Background       | Camera           | Detailed        | Main Object      | Short            | Token/Sec   | Peak Memory Overhead   |
> |-----------------------------|----------|------------------|------------------|-----------------|------------------|------------------|---------|---------|
> | Llava-Next-Video-7B-DPO@AuroraCap     | Vanilla  | 38.55 / 2.0008   | 37.68 / 1.951    | 42.91 / 2.2238  | 40.88 / 2.0954   | 41.63 / 2.1500   |  27  | 62.89 GB   |
> |                             | Ours     | 38.50 / 1.9905    | 37.61 / 1.9353    | 42.53 / 2.2036   | 40.97 / 2.1136   | 41.62 / 2.1486    |  34  |   25.04GB   |

---

> ### Author Response · Authors · 2024-11-25
>
> **Q1**: Reusing KV Caches When Resuming the Generation Process
>
> **AQ1**: Yes, **the KV cache of the previously generated video tokens (i.e., $P_C$) can indeed be reused**. The only additional overhead arises from the need to regenerate previously generated output tokens, as these cannot be directly reused. However, the KV cache for video tokens remains fully preserved and reusable, minimizing the computational load associated with resuming the generation process.
>
> **Q2**: Why duplicating tokens is an effective approach
>
> **AQ2**: This is an excellent question! The effectiveness of duplicating tokens lies in its ability to **guide the VLM’s attention**. Given the strong feature extraction capabilities of VLMs, our insight is that **the key to improving performance is not necessarily introducing additional information but rather emphasizing the importance of specific information**. By duplicating tokens, we aim to effectively highlight the significance of the identified important tokens, steering the model’s attention toward them during processing.
>
> **Reference**:
>
> [1] OpenAI. (2024). Migrating to replacements. Retrieved November 25, 2024, from https://platform.openai.com/docs/deprecations#migrating-to-replacements:~:text=RECOMMENDED%20REPLACEMENT-,2024%2D09%2D13,gpt%2D3.5%2Dturbo,-Fine%2Dtuned%20models.
>
> [2] Wang, Xidong, et al. "LongLLaVA: Scaling Multi-modal LLMs to 1000 Images Efficiently via a Hybrid Architecture." arXiv preprint arXiv:2409.02889 (2024).
>
> [3] Xu, Mingze, et al. "Slowfast-llava: A strong training-free baseline for video large language models." arXiv preprint arXiv:2407.15841 (2024).
>
> [4] Zhang, Ruohong, et al. "Improve Vision Language Model Chain-of-thought Reasoning." arXiv preprint arXiv:2410.16198 (2024).
>
> [5] Zhou, Junjie, et al. "MLVU: A Comprehensive Benchmark for Multi-Task Long Video Understanding." arXiv preprint arXiv:2406.04264 (2024).
>
> [6] Chai, Wenhao, et al. "AuroraCap: Efficient, Performant Video Detailed Captioning and a New Benchmark." arXiv preprint arXiv:2410.03051 (2024).

---

> > ### Comment · Reviewer_QnQy · 2024-12-02
> >
> > Thank you to the authors for their efforts. Most of my concerns have been addressed. I will raise my rating to 5.

---

> > > ### Author Response · Authors · 2024-12-02
> > >
> > > Dear Reviewer QnQy,
> > >
> > > Happy Thanksgiving! Thank you for recognizing that our response has addressed most of your concerns!
> > >
> > > If you have any additional comments or concerns, please don’t hesitate to let us know. We would be more than happy to address them!

---

### Official Review · Reviewer_hSzd · 2024-10-28

**Soundness:** 3
**Presentation:** 3
**Contribution:** 3
**Rating:** 5
**Confidence:** 4

**Summary:**

This work focuses on efficient video processing framework for video VLMs. It proposes a pipeline to first generates a high-level overview of the entire video and then adaptively zooms in on specific parts based on the content being generated. Experiments show effectiveness on the video detailed description dataset.

**Strengths:**

1. The efficient video compression for VLM is promising and useful for practical usage.
2. The course-to-fine design for video understanding is interesting and seems to be useful in caption.
3. Overall, the writing is clear and easy to follow.

**Weaknesses:**

1. From Figure 2, the proposed ZoomVLM utilizes LLM twice for image-level and token level selection. But the efficency in Table 1 is seems better for vanilla manner in Token/sec. Is the efficiency mainly from reduction in video tokens? Considering provide the whole inference time and time spent on each component (video overview generation, token adjustment, etc.) for clear comparison. It could be better to compare other metrics like memory usage or FLOPs.
2. The evalution in Video description is far from enough. The authors are recommended to conduct experiments on VideoMME [A] and EgoSchema [B], which could be more close to real-life scenes. Of course, it's better to provide some analysis or limitation on different benchmarks.
3. One of the main drawback in current pipeline could be the multi-round QA, which is more useful in practical applications. Because the Vanilla or Slowfast do no need to generate the token again for different round. Are the authors have any solutions or ideas to this tasks or potential optimizations for repeated queries on the same video?
4. Because the video overview augmenter and adaptive token adjustment are all target to reduce reduent tokens, why not only keep the adaptive token adjustment with larger reduction rate? The authors are recommended to discuss any potential synergies or trade-offs between the two approaches.

[A] "Video-MME: The First-Ever Comprehensive Evaluation Benchmark of Multi-modal LLMs in Video Analysis", arXiv:2405.21075, 2024

[B] "Egoschema: A diagnostic benchmark for very long-form video language understanding", NeurIPS, 2023

**Questions:**

My main concern focuses on the experiment part. Current experiments are not enought to support the general efficient framework for VLM.

---

> ### Author Response · Authors · 2024-11-25
> **Response to Reviewer hSzd**
>
> We greatly appreciate your review efforts. Thank you for your encouraging recognition of **ZoomVLM's research is promising and useful, our coarse-to-fine design is interesting and useful, and our presentation is clear and easy to follow**! Below, we address your questions/comments and provide detailed clarifications:
>
> **W1**: Source of Efficiency and further illustration on sources of efficiency and metrics.
>
> **A1**: Yes, your understanding is correct. The **primary source of ZoomVLM's efficiency stems from the reduced number of video tokens**. To provide further clarity on the sources and metrics of efficiency, we have conducted detailed profiling and theoretical analysis. The results are as follows:
>
> > (1) **Profiling Results**: The table below presents the profiling results for each module in ZoomVLM, measured on the state-of-the-art VLM, Llava-Next-Video-7B_DPO, during inference on the VDD dataset [1]. The output consists of 200 tokens, similar to the average output length in the VDD dataset. The additional overhead introduced by the **Video Overview Augmenter and Adaptive Token Adjustment accounts for approximately 5% and 1% of the total inference cost, respectively**. Despite this, **the improved token efficiency introduced by these two modules leads to a ~25% reduction in the total inference cost**, sourced from ~80% less latency in the prefilling stage and ~20% reduction in latency of auto-regressive generation due to a ~40% fewer tokens needed to represent video. We have included these profiling results in the appendix of our manuscript.
> | Model     | Video Overview Augmenter (s) | Adaptive Token Adjustment (s) | Backbone Inference (Autoregressive) | Backbone Inference (Prefill) | Total Inference Time | VDD Score |
> |-----------|------------------------------|--------------------------------|--------------------------------------|-------------------------------|----------------------|-----------|
> | Vanilla   | 0                            | 0                              | 7.8338                               | 1.0812                        | 8.915                | 2.843     |
> | ZoomVLM   | 0.3277                       | 0.0705                         | 6.279                                | 0.2215                        | 6.8987               | 3.102     |
>
>
>
> > (2) **Memory Efficiency**: Following your suggestion, we have also analyzed the peak memory reduction achieved by ZoomVLM, which results from the reduced number of video tokens. Specifically, we generated 300 tokens using different VLMs, measured their peak memory usage, and summarized the results in the table below. The results show that **ZoomVLM reduces peak memory usage by 29% to 32% compared to the most competitive baseline (i.e., Slowfast-llava [2])**. This set of experiments further highlights the comprehensive efficiency improvements of ZoomVLM, covering both latency and peak memory overhead. We have included this result in the appendix of our manuscript.
> | Models                  | Method      | Peak Memory (300 tokens) |
> |-------------------------|-------------|---------------------------|
> | Llava-Next-Video-7B-DPO | Vanilla     | 62.89GB                  |
> |                         | Slow-Fast   | 36.46GB                  |
> |                         | Ours        | 25.04GB                  |
> | llava-v1.6-vicuna-13b   | Vanilla     | OOM                      |
> |                         | Slow-Fast   | 60.4GB                   |
> |                         | Ours        | 42.95GB                  |

---

> ### Author Response · Authors · 2024-11-25
>
> **W2**: Analysis of different benchmarks and evaluation of more benchmarks
>
> **A2**: First, we would like to clarify that ZoomVLM is designed to improve the efficiency of Video VLMs during **open-ended generation tasks**, where the model generates a sequence of tokens to comprehensively answer an open-ended question. This focus is motivated by two key reasons:
>
> > (1) **Efficiency Bottleneck in Open-Ended Generation**: Open-ended generation faces significant efficiency challenges [3, 4, 5], as highlighted in the profiling results provided in **A1**. Specifically, prefill operations in vanilla VLM inference consume less than 20% of the total inference cost, whereas the autoregressive generation process dominates the remaining cost, increasing proportionally with the generated context length [3, 4, 5]. Addressing this bottleneck is critical for improving overall efficiency.
>
> > (2) **Relevance to Real-World Applications**: Open-ended generation tasks are more prevalent in real-world scenarios compared to multiple-choice or word-level generation tasks. They allow VLMs to produce comprehensive and nuanced responses, providing a more meaningful evaluation of the model’s performance [6, 7, 8].
>
> Furthermore, following your suggestion, we conducted additional experiments to evaluate ZoomVLM on a broader range of benchmarks, including **MLVU [9], which focuses on long video understanding tasks, and AuroraCap [10]**. The results, summarized in the table below, demonstrate that ZoomVLM consistently achieves comparable or superior accuracy while delivering improved efficiency compared to the baseline solutions. This set of experiments further validates the general effectiveness and applicability of ZoomVLM across diverse video understanding tasks. We have included these experiments in the appendix of our manuscript.
>
> | Setting                      | Method   | SSC     | VS      | G-Avg   | Token/Sec   | Peak Memory Overhead   |
> |-----------------------------|----------|---------|---------|---------|---------|---------|
> | Llava-Next-Video-7B-DPO@MLVU     | Vanilla  | 3.5743  | 2.6523  | 3.1133  |  25    | 62.89 GB   |
> |                             | Ours     | 3.5095  | 2.6714  | 3.09045 |  32  |   25.04GB   |
>
>
> | Model                       | Method   | Background       | Camera           | Detailed        | Main Object      | Short            | Token/Sec   | Peak Memory Overhead   |
> |-----------------------------|----------|------------------|------------------|-----------------|------------------|------------------|---------|---------|
> | Llava-Next-Video-7B-DPO@AuroraCap     | Vanilla  | 38.55 / 2.0008   | 37.68 / 1.951    | 42.91 / 2.2238  | 40.88 / 2.0954   | 41.63 / 2.1500   |  27  | 62.89 GB   |
> |                             | Ours     | 38.50 / 1.9905    | 37.61 / 1.9353    | 42.53 / 2.2036   | 40.97 / 2.1136   | 41.62 / 2.1486    |  34  |   25.04GB   |
>
> **W3**: Support for multi-round QA
>
> **A3**: Thank you for this interesting question! We would like to clarify that **ZoomVLM seamlessly supports multi-round QA tasks**. Specifically, the Video Overview Augmenter and Adaptive Token Adjustment modules in ZoomVLM convert the video representation into a question-specific format, and **this transformation can be reverted with a small cost (i.e., less than 6% of the total inference overhead)**. For subsequent rounds of QA with the same video, we propose the following process:
>
> > (1) **Regeneration of the Video Overview**: For a new question, the Video Overview Augmenter can regenerate a video overview specific to the query. As demonstrated in our provided profiling results in our response to W1, this step incurs only ~5% additional overhead compared to vanilla inference, while reducing the number of video tokens by nearly 40%.
>
> > (2) **Resetting the Video Token Representation**: The video token representation can be efficiently reset to the previously generated video overview by leveraging a cached set of dropped tokens. This operation introduces a negligible additional overhead (i.e., less than 1%, as shown in the profiling results in **A1**).
>
> Moreover, our observations indicate that as long as the generated video overview approximately preserves the information in the video, Adaptive Token Adjustment can significantly recover performance. This opens up the possibility of reusing an already generated video overview without regenerating a question-specific overview for each round in a multi-round QA setting.
> As shown in the table below, even when using a video overview generated via random frame selection, applying Adaptive Token Adjustment can largely recover performance compared to using an accurate video overview as in the full ZoomVLM pipeline.
> | Method                | Random | Random + Adjustment | ZoomVLM |
> |-----------------------|--------|----------------------|---------|
> | Video_DC             | 2.908  | 2.986               | 3.102   |

---

> ### Author Response · Authors · 2024-11-25
>
> **W4**: Why not only keep the adaptive token adjustment with a larger reduction rate
>
> **A4**: Thank you for this insightful suggestion! One of the key insights we gained in improving token efficiency during inference is that while **moderate adjustments to the video token representation can improve the performance of VLMs, drastic changes often result in significant performance degradation**. We hypothesize that this occurs because large-scale changes to the KV cache can disrupt the internal consistency of the VLM, leading to suboptimal or meaningless outputs.
>
> To validate this hypothesis, we conducted additional experiments using only the Adaptive Token Adjustment module to reduce the number of video tokens with varying reduction rates. As shown in the table below, **slight adjustments** (e.g., dropping and copying fewer than 30 tokens per adjustment) improved the VDD score (e.g., from 0.006 to 0.032). However, more **aggressive adjustments** led to a significant performance drop (e.g., from 0.102 to 0.909), confirming that extreme token reductions negatively impact model performance. Thus, it is critical to first leverage the Video Overview Augmenter to generate a video overview and largely reduce the number of video tokens, then introduce the Adaptive Token Adjustment to calibrate video representation and further improve the response accuracy. We have included this experiment in the appendix of our manuscript.
>
> | Total # of Reduced Tokens | 0      | 15     | 30    | 75     | 105    | 150    | 300    |
> |----------------------|--------|--------|-------|--------|--------|--------|--------|
> | # of Dropped Tokens | 0      | 10     | 20    | 50     | 70     | 100    | 200    |
> | # of Copied Tokens   | 0      | 5      | 10    | 25     | 35     | 50     | 100    |
> | VDD Score           | 2.843  | 2.849 | 2.875   | 2.741 | 2.386 | 2.148 | 1.934 |
>
>
> **Reference:**
>
> 1] Li, Feng, et al. "Llava-next-interleave: Tackling multi-image, video, and 3d in large multimodal models." arXiv preprint arXiv:2407.07895 (2024).
>
> [2] Xu, Mingze, et al. "Slowfast-llava: A strong training-free baseline for video large language models." arXiv preprint arXiv:2407.15841 (2024).
>
> [3] Wu, Wei, et al. "TokenSelect: Efficient Long-Context Inference and Length Extrapolation for LLMs via Dynamic Token-Level KV Cache Selection." arXiv preprint arXiv:2411.02886 (2024).
>
> [4] Xiao, Guangxuan, et al. "DuoAttention: Efficient Long-Context LLM Inference with Retrieval and Streaming Heads." arXiv preprint arXiv:2410.10819 (2024).
>
> [5] Xiao, Guangxuan, et al. "Efficient streaming language models with attention sinks." arXiv preprint arXiv:2309.17453 (2023).
>
> [6] Ging, Simon, María A. Bravo, and Thomas Brox. "Open-ended VQA benchmarking of Vision-Language models by exploiting Classification datasets and their semantic hierarchy." arXiv preprint arXiv:2402.07270 (2024).
>
> [7] Krishna, Kalpesh, Aurko Roy, and Mohit Iyyer. "Hurdles to progress in long-form question answering." arXiv preprint arXiv:2103.06332 (2021).
>
> [8] Maaz, Muhammad, et al. "Video-chatgpt: Towards detailed video understanding via large vision and language models." arXiv preprint arXiv:2306.05424 (2023).
>
> [9] Zhou, Junjie, et al. "MLVU: A Comprehensive Benchmark for Multi-Task Long Video Understanding." arXiv preprint arXiv:2406.04264 (2024).
>
> [10] Chai, Wenhao, et al. "AuroraCap: Efficient, Performant Video Detailed Captioning and a New Benchmark." arXiv preprint arXiv:2410.03051 (2024).

---

> > ### Author Response · Authors · 2024-12-02
> >
> > Dear Reviewer hSzd,
> >
> > Happy Thanksgiving! Thank you for taking the time to provide your constructive feedback on our paper.
> >
> > As the reviewer-author discussion period approaches its deadline (Dec. 2, AoE), we look forward to hearing any additional comments or concerns you may have. We are happy to address any points to further clarify or improve our work.
> >
> > Thank you again for your valuable insights, and we look forward to your feedback!

---

### Official Review · Reviewer_vk7p · 2024-11-02

**Soundness:** 2
**Presentation:** 3
**Contribution:** 2
**Rating:** 5
**Confidence:** 5

**Summary:**

This paper presents ZoomVLM, a novel framework aimed at enhancing the efficiency of vision-language models (VLMs) for video understanding. The authors address a critical issue: existing VLMs, particularly SOTA models like Llava-OneVision, require a high number of tokens, leading to slow inference and computational bottlenecks. Inspired by human perception strategies--where people focus on general overviews and selectively zoom into specific areas for details--ZoomVLM proposes a more selective token allocation approach to reduce token usage while maintaining performance.

**Strengths:**

1. Tuning-free and plug-and-play: Being tuning-free and plug-and-play, ZoomVLM can be seamlessly integrated with existing VLMs without the need for extensive modifications or retraining, facilitating broader adoption.
2. Efficient Token Usage. Selectively allocating tokens to the most important parts of a video is an intuitive motivation.
3. Well-structured presentation. The presentation of this paper is clear and easy to understand.

**Weaknesses:**

1. The motivation is somewhat unclear. The generation quality of Video Overview Augmenter is crucial, as it determines which parts of the video should be emphasized or ignored. However, due to spatial pooling and temporal sampling, the quality may be suboptimal, and this has not been validated through ablation studies.
2. There are concerns about its practicality. First, the Video Overview Augmenter increases inference costs for the same question compared to other methods. Second, the proposed method only achieves comparable performance compared to its counterparts.
3. Lack sufficient experimental support. It would be beneficial to include evaluations on other challenging video benchmarks, such as long video datasets, to validate effectiveness and enable a more comprehensive comparison.

**Questions:**

See Weaknesses.

---

> ### Author Response · Authors · 2024-11-25
> **Response to Reviewer vk7p**
>
> We greatly appreciate your review efforts. Thank you for your encouraging recognition of ZoomVLM's **tuning-free, plug-and-play design, efficient token usage, and well-structured presentation**! Below, we address your questions/comments and provide detailed clarifications:
>
>
> **W1**: Motivation on Video Overview Augmenter and validation on the optimality of the generated video overview.
>
> A1: Thank you for highlighting this important aspect of our work! The quality of the video overview generated by the Video Overview Augmenter is critical, as it directly influences the information available to the Vision-Language Model (VLM) for generating accurate responses. To address your question and validate the effectiveness of our approach in generating the video overview, we conducted the following experiments:
>
> > (1) **Comparison of VLM Performance with Original Videos vs. Generated Overviews**: Table 3 in our paper (replicated below) compares the performance of the VLM when using the original video versus the generated video overview as input to quantify the potential information loss in the video overview. Notably, the video overview generated by our Video Overview Augmenter (referred to as "Summary Only") achieves comparable VLM response quality to the original video, despite requiring approximately 40% fewer video tokens.
> | Setting                          | Original Video| Summary Only |
> |-----------------------------------|---------|----------------------------------|
> | # Tokens                | 4608           | 2848                      |
> | VDD Score                        | 2.843   | 2.801                            |
>
>
> > (2) **Comparison of Keyframe Selection Methods**: Table 5 in our paper (replicated below) evaluates the effectiveness of our Video Overview Augmenter's keyframe selection strategy against alternative approaches such as random sampling and uniform sampling, as used in prior works [1]. Our method achieves a 0.082~0.116 higher VDD score than baseline solutions.
> | Selection Method | Random Sample | Uniform Sample | Ours  |
> |-------------------|--------|---------|-------|
> | Score             | 2.986  | 3.020    | 3.102 |
>
>
> > (3) **Validation of the Video Overview Format**: We further examined the efficacy of our chosen video overview format, which integrates a high-level overview with a few keyframes. As shown in the table below, our approach outperforms commonly used methods such as spatial pooling only and temporal sampling only [2, 3, 4] in terms of VDD score, while maintaining comparable token efficiency. We have added this experiment to the appendix of our manuscript.
> | Setting                 | Original Video | Spatial Pooling | Temporal Sampling | Video Overview Augmenter |
> |--------------------------|----------------|------------------|--------------------|---------------------------|
> | # Tokens                | 4608           | 2048             | 2880              | 2848                      |
> | VDD Score               | 2.843          | 2.346            | 2.727             | 2.801                     |

---

> ### Author Response · Authors · 2024-11-25
>
> **W2**: Concerns about the practicality of ZoomVLM because (1) Video Overview Augmenter increases inference costs and (2) ZoomVLM achieves comparable performance as baseline.
>
> **A2**: Thank you for sharing your concerns. Below, we address both points in detail:
>
> > (1) **Inference Costs of the Video Overview Augmenter**: The additional overhead introduced by the Video Overview Augmenter is trivial compared to the token efficiency it enables. As demonstrated in the profiling table below, the **Video Overview Augmenter accounts for only ~5% of the total generation latency** when producing an output of 300 tokens (for context, the average output length in VDD is approximately 300 tokens). This marginal cost is a small trade-off considering the significant reduction in the token usage it achieved. We have added the profiling results to the appendix of our manuscript.
> | Dataset   | Model                     | Video Overview Augmenter | Adaptive Token Adjustment | Backbone Inference (Autoregressive) | Backbone Inference (Prefill) | Total Inference time |
> |---------|-----------|---------------------------|------------------------------|--------------------------------|--------------------------------------|-------------------------------|
> | Video_DC  | Llava-Next-Video-7B-DPO   | 0.3277                       |0.0705                         | 6.279                                | 0.2215                        | 6.8987               |
>
> > (2) **Concerns About the Performance of ZoomVLM**: We humbly clarify that the improvements achieved by ZoomVLM are not merely comparable but quite significant. Specifically, ZoomVLM achieves **up to 30% lower latency alongside a 0.259 improvement in the VDD score** compared to baseline solutions. This level of improvement, particularly in a tuning-free, plug-and-play framework, is substantial. As references, recent works that are highly cited and/or published in top-tier conferences that aim to improve LLM efficiency typically require model tuning and achieve efficiency gains of less than 30% [5, 6, 7, 8]. The fact that ZoomVLM delivers both higher efficiency and comparable or better accuracy without requiring any model tuning demonstrates its practicality and significance making it a nontrivial contribution to the field in our humble opinion.
>
> **W3**: Lack of experimental support
>
> **A3**: Thank you for suggesting extra evaluations which we believe can help strengthen our work! Following your suggestion, we conducted additional experiments on a broader range of benchmarks, including MLVU [9], which focuses on long video understanding tasks, and AuroraCap [10]. As shown in the table below, ZoomVLM consistently achieves comparable or superior accuracy while demonstrating improved efficiency compared to the baseline solution. These results further validate the generalizability of ZoomVLM across diverse benchmarks. We have added these experiments to the appendix of our manuscript.
>
> | Setting                      | Method   | SSC     | VS      | G-Avg   | Token/Sec   | Peak Memory Overhead  (300 Tokens)  |
> |-----------------------------|----------|---------|---------|---------|---------|---------|
> | Llava-Next-Video-7B-DPO@MLVU     | Vanilla  | 3.5743  | 2.6523  | 3.1133  |  25    | 62.89 GB   |
> |                             | Ours     | 3.5095  | 2.6714  | 3.09045 |  32  |   25.04GB   |
>
>
> | Model                       | Method   | Background       | Camera           | Detailed        | Main Object      | Short            | Token/Sec   | Peak Memory Overhead (300 Tokens)  |
> |-----------------------------|----------|------------------|------------------|-----------------|------------------|------------------|---------|---------|
> | Llava-Next-Video-7B-DPO@AuroraCap     | Vanilla  | 38.55 / 2.0008   | 37.68 / 1.951    | 42.91 / 2.2238  | 40.88 / 2.0954   | 41.63 / 2.1500   |  27  | 62.89 GB   |
> |                             | Ours     | 38.50 / 1.9905    | 37.61 / 1.9353    | 42.53 / 2.2036   | 40.97 / 2.1136   | 41.62 / 2.1486    |  34  |   25.04GB   |

---

> > ### Author Response · Authors · 2024-11-25
> >
> > **References**:
> >
> > [1] Xu, Mingze, et al. "Slowfast-llava: A strong training-free baseline for video large language models." arXiv preprint arXiv:2407.15841 (2024).
> >
> > [2] Wu, Yecheng, et al. "Vila-u: a unified foundation model integrating visual understanding and generation." arXiv preprint arXiv:2409.04429 (2024).
> >
> > [3] Lin, Bin, et al. "Video-llava: Learning united visual representation by alignment before projection." arXiv preprint arXiv:2311.10122 (2023).
> >
> > [4] Li, Bo, et al. "Llava-onevision: Easy visual task transfer." arXiv preprint arXiv:2408.03326 (2024).
> >
> > [5] Ma, Xinyin, Gongfan Fang, and Xinchao Wang. "Llm-pruner: On the structural pruning of large language models." Advances in neural information processing systems 36 (2023): 21702-21720.
> >
> > [6] Zhao, Bowen, Hannaneh Hajishirzi, and Qingqing Cao. "Apt: Adaptive pruning and tuning pretrained language models for efficient training and inference." arXiv preprint arXiv:2401.12200 (2024).
> >
> > [7] Kurtić, Eldar, Elias Frantar, and Dan Alistarh. "Ziplm: Inference-aware structured pruning of language models." Advances in Neural Information Processing Systems 36 (2024).
> >
> > [8] Men, Xin, et al. "Shortgpt: Layers in large language models are more redundant than you expect." arXiv preprint arXiv:2403.03853 (2024).
> >
> > [9] Zhou, Junjie, et al. "MLVU: A Comprehensive Benchmark for Multi-Task Long Video Understanding." arXiv preprint arXiv:2406.04264 (2024).
> >
> > [10] Chai, Wenhao, et al. "AuroraCap: Efficient, Performant Video Detailed Captioning and a New Benchmark." arXiv preprint arXiv:2410.03051 (2024).

---

> > ### Comment · Reviewer_vk7p · 2024-11-26
> >
> > Thank you for your rebuttal! While some of my concerns have been adequately addressed, issues related to long-video scenarios remain unresolved. Additionally, the experimental results are not convincing. AuroraCap does not appear to be a suitable benchmark for long videos, and the results on MLVU do not demonstrate a significant improvement over the baseline.

---

> ### Author Response · Authors · 2024-11-26
>
> Thank you for your prompt response! We are glad to hear that some of your concerns have been addressed and we appreciate the opportunity to clarify further.
>
> Regarding long-video scenarios, we have included two efficiency metrics in the MLVU results to address this concern: the number of generated tokens per second (Token/Sec) and the peak memory overhead. **ZoomVLM shows a 26% improvement in Token/Sec and a 60% reduction in peak memory overhead compared to the baseline vanilla model**, while maintaining comparable accuracy. Considering the training-free and plug-and-play nature of ZoomVLM, we would like to humbly emphasize that ZoomVLM's improvements are meaningful and represent a significant advancement in addressing efficiency challenges for long video scenarios.
>
> To further address your concern, we are currently evaluating additional models on the MLVU benchmark and will share the results as soon as they are available. In the meantime, if you have specific suggestions for experiments or additional explanations you would like us to provide, we would be happy to consider them to address your concerns more thoroughly!

---

> ### Author Response · Authors · 2024-12-02
>
> Dear Reviewer vk7p,
>
> Happy Thanksgiving! Thank you for your constructive review and for engaging in discussions with us during the rebuttal period.
>
> As promised, we conducted additional experiments on the MLVU long video understanding benchmark with more VLMs to further validate ZoomVLM's performance in handling long videos. The results, summarized below, show that ZoomVLM achieves **a 0.053\~0.057 higher G-Avg score and a 25.0%\~27.3% higher generation speed (tokens/sec)** compared to the vanilla implementation on the **newly evaluated Llava-Next-Video-7B and Llava-Next-Interleave-7B-DPO models**. These findings demonstrate ZoomVLM's ability to enhance generation efficiency while maintaining, if not improving, its video understanding capabilities.
>
> If you have any additional concerns about ZoomVLM's ability to preserve the original VLM's long video understanding performance or any other aspects, please feel free to let us know. We would be more than happy to address them!
>
> | Model                              | Method   | SSC   | VS    | G-Avg | Token/sec |
> |------------------------------------|----------|-------|-------|-------|-----------|
> | Llava-Next-Video-7B-DPO            | Vanilla  | 3.574 | 2.652 | 3.113 | 27        |
> |                                    | Ours     | 3.509 | 2.671 | 3.09  | 31        |
> | Llava-Next-Video-7B                | Vanilla  | 3.143 | 2.573 | 2.858 | 22        |
> |                                    | Ours     | 3.143 | 2.687 | 2.915 | 28        |
> | Llava-Next-Interleave-7B-DPO       | Vanilla  | 3.971 | 2.134 | 3.053 | 16        |
> |                                    | Ours     | 3.895 | 2.316 | 3.106 | 20        |

---

### Meta-Review · Area_Chair_XygQ · 2024-12-19

**Metareview:**

This paper proposed a novel framework named ZoomVLM to improve the efficiency of VLM by reducing the number of video tokens. ZoomVLM contains two key components: (1) A Video Overview Augmenter which creates a video summary; (2) An Adaptive Token Adjustment which predicts the significance of different video parts for video token allocation. The proposed ZoomVLM is tuning-free and could be used as plug-and-play video processing pipeline for VLM. Experiments on several VLM models show improvement of efficiency by applying the proposed ZoomVLM.

Strength:
1. As agreed by reviewers, the proposed method is tuning-free and could be used in the way of plug-and-play.
2. The proposed method is promising to improve the efficiency of VLM.

Weakness:
The major concerns proposed by reviewers are:
1. Experiments are not sufficient (mentioned by all reviewers), especially for long video understanding and other video understanding tasks such as VideoMME and EgoSchema. The authors replied to these concerns during discussion, however, the concerns are not fully addressed based on the reviewers' feedback. The authors claimed that the paper target for open-ended video understanding tasks, however, tasks as proposed in VideoMME and EgoSchema are also important tasks of video understanding. Studies on broad areas of video understanding would make the paper solid.

2. Two of the reviewers mentioned that the proposed two components also takes time, especially for the Video Overview Augmenter which includes LLM to generate a video summary. The authors answered these questions in the discussion by showing that the runtime of the proposed components is a small portion of the total runtime. This partially addressed the concerns raised by the reviewers. However, as reviewer hSzd mentioned, comparing to other methods based on FLOPS is better. I agree with hSzd as the runtime may depend on the implementation and infrastructure, FLOPS is a better way to show the efficiency and would probably reduce some concerns raised by the reviewers.

**Additional Comments On Reviewer Discussion:**

All reviewers mentioned that the experiments are sufficient. The authors try to address this by providing more results, however, this still cannot address the concerns from the reviewers. This is the major concern about this paper. The authors may want to include more thorough experiments and include more tasks in video understanding to make the paper solid.

Two of the three reviewers show their concerns about the additional cost introduced by the new components, especially the Video Overview Augmenter makes use of LLM. The authors tried to address these concerns by showing that the runtime of the new components are negligible. This addresses the concerns. However showing FLOPS as suggested by hSzd may reduce the concerns from the reviewers.

---

### Decision · Program_Chairs · 2025-01-22

Reject